

# Assessment of the contribution of IRS for the characterisation of ozone over Europe

Francesca Vittorioso[1], Vincent Guidard[1], and Nadia Fourrié[1]

[1]CNRM, Université de Toulouse, Météo-France, CNRS, Toulouse, France

**Correspondence:** francesca.vittorioso@outlook.com

**Abstract.** In the coming years, EUMETSAT's Meteosat Third Generation - S (MTG-S) satellites will be launched with an instrument of valuable features on board. The MTG - Infrared Sounder (IRS) will represent a major innovation for the monitoring of the chemical state of the atmosphere, since, at present, observations of these parameters mainly come from *in situ* measurements (geographically uneven) and from instruments on board of polar-orbiting satellites (highly dependent on the

scanning line of the satellite itself, which is limited, over a specific geographical area, to very few times per day). MTG-IRS will present many potentials in the area of detecting different atmospheric species and will have the advantage of being based on a geostationary platform and to acquire data with a high temporal frequency (every 30 minutes over Europe), which makes easier to track the transport of the species of interest.

The present work aims to evaluate the potential impact, over a regional domain over Europe, of the assimilation of IRS

radiances within a chemical transport model (CTM) Modèle de Chimie Atmosphérique de Grande Echelle (MOCAGE), operational in Météo-France.

Since IRS is not yet in orbit, observations have been simulated using the Observing System Simulation Experiment (OSSE) approach. Of the species to which IRS will be sensitive, the one treated along this study was the ozone.

The results obtained indicate that the assimilation of synthetic radiances of IRS always has a positive impact on the ozone

analysis from the model MOCAGE. The relative average difference compared to the NR in the ozone total columns improves from -30% (no assimilation) to almost zero when IRS observations are available over the domain. When considering tropospheric columns the improvement is also significant, from 15-20% (no assimilation) down to 3%.

## 1  Introduction

Many efforts are made in research to ensure an accurate monitoring of the atmospheric chemical state. This is necessary in order to be able, whenever necessary, to take the right steps to rectify unhealthy behaviours or to take measures to protect environment and population health. To do so, Chemistry Transport Models (CTMs) that predict this behaviour with quality and



precision are required, with a specific focus on both the efficiency of codes and methods themselves, and on their integration with real observations of the atmospheric state.

At present the observation system of the atmospheric chemical composition is mainly based on *in situ* measurements and observations from polar-orbiting satellites. The *in situ* measurements can concern surface observations, carried out through ground-based measurement stations, or airborne observations, acquired trough instruments boarded on aircrafts, balloons or drones. Although this kind of observations prove to be extremely helpful for atmospheric sciences, they do not exist everywhere, since huge gaps exist between weather stations.

Measurements from polar-orbiting satellites, on the other hand, cover much more land surface on a regular basis. Among the most efficiently used tools belonging to this category, definitely worth mentioning is the Infrared Atmospheric Sounding Interferometer (IASI), one of the main payloads of the polar-orbiting Meteorological Operational Satellite (Metop) series, developed by ESA and operated by the EUMETSAT agency [Siméoni et al. (1997); Blumstein et al. (2004)]. IASI provides accurate measurements of the meteorological and chemical state of the Earth's atmosphere. The instrument acquires spectra of

atmospheric emission within 645 and 2760 $cm^{-1}$, with a spectral apodized resolution of 0.5 $cm^{-1}$ and a 0.25 $cm^{-1}$ spectral sampling. Consequently, it measures at 8461 wavelengths (or channels). Thanks to its fine spectral resolution, signal-to-noise ratio and wide spectrum range, it is a precious resource for detecting trace gases like ozone, methane and carbon monoxide, as well as clouds, aerosols and greenhouse gases [Phulpin et al. (2002); Clerbaux et al. (2009); Hilton et al. (2012); Barret et al. (2020)].

However, satellite instruments in polar orbit are able to provide repeated measurements on the same point on Earth only a few times during the same day. Having IASI-like measurements coming from instruments on board a geostationary (GEO) platform has the potential to provide significant advantages in the area of atmospheric chemistry monitoring. These would acquire much wider views of Earth with a much higher temporal acquisition frequency.

     Geostationary motions require the satellite to cover an orbit much wider and further from the surface than satellites in polar

orbit. This has been for years at the expense of both spatial and spectral resolution of the acquired data. However, the enormous technological developments of the recent decades allowed to develop geo-instruments producing a high-resolution spectral information close to that obtainable from a polar instrument.

     For this reason only very recently IASI-like satellite instruments for the study of chemistry monitoring are appearing on board of geostationary platforms. The Chinese Geostationary Interferometric Infrared Sounder (GIIRS) is the first hyperspectral

infrared sounder on board a geostationary satellite, namely the FengYun-4 series launched in 2016 (FY-4A) and 2021 (FY-4B) [Yang et al. (2017)]. With its hyperspectral coverage [i.e. from 680 to 1130 $cm^{-1}$ and from 1650 to 2250 $cm^{-1}$] with a spectral resolution of 0.625 $cm^{-1}$, it is able to provide important data for monitoring CO [Zeng et al. (2022)] or, for instance, the $NH_3$ cycle [Clarisse et al. (2021)]. GIIRS, however, is focused on the monitoring of a limited area over Asia.

     The European agency EUMETSAT, on the other hand, has also envisaged to put an infrared sounder on board the geosta-

tionary Meteosat Third Generation - Sounding (MTG-S) scheduled to be launched as part of the MTG satellite series. This instrument, designed and built by Thales Alenia Space, is the Infrared Sounder (IRS) indeed. This will provide data of the full Earth disk with an acquisition every 30 minutes over Europe. Although initially designed for meteorological observations,



IRS has ample potential for its exploitation in atmospheric chemistry monitoring. It will acquire spectra in a long wavelength infrared (LWIR) $[679.70 - 1210.44 \, \text{cm}^{-1}]$ and a medium wavelength infrared (MWIR) $[1600.00 - 2250.20 \, \text{cm}^{-1}]$ spectral band with a 0.603 and 0.604 $\text{cm}^{-1}$ spectral sampling respectively[1].

This research work is among the preliminary studies being carried out to prepare for the arrival of this powerful tool. The main object of these authors has been to assess the contribution of the data that MTG-IRS will soon provide for the characterisation of the atmospheric chemical composition over Europe through the assimilation of these data into a CTM. Such evaluation has been carried out by simulating IRS Level 1 (L1) data first, and then performing their assimilation into MOCAGE[2].

Since this study is a preliminary study, the focus is put on a single species among those that the instrument will be able to detect and which, on the other hand, plays an important role in the unfolding of many atmospheric processes, namely the ozone.

The latter, indeed, is a trace gas and a secondary species (i.e. not emitted) resulting from photochemical reactions. It is mainly found in stratosphere (almost 90%), resulting from photodissociation of oxygen. The stratospheric ozone layer is of paramount importance in shielding harmful ultraviolet solar radiation and it makes possible life on Earth. On the other hand, a smaller percentage of ozone is found in troposphere. It comes in small part from the stratosphere itself and in larger amounts from reactions in the lower layers involving primary compounds of natural or anthropogenic origin. Tropospheric ozone can cause issues to human health and the ecosystem, as well as to agriculture and material goods due to its high oxidant power.

In the course of this manuscript, the characteristics of IRS (Section 2), those of the CTM MOCAGE (Section 3) and the data assimilation methods chosen and employed (Section 4) will be described in more detail. An extensive space will be, then, devoted in Section 5 to the necessary description of the construction of an OSSE. The choice to focus on the ozone species and to simulate and assimilate L1 data will be better justified. In Section 6 he results of the research will be presented, while conclusions and perspectives will be provided in the last section.

## 2  MTG Infrared Sounder (IRS)

The information about IRS reported in this subsection are issued from EUMETSAT, Thales Alenia Space and Coopmann et al. (2022).

IRS is a Fourier Transform Spectrometer, built by Thales Alenia Space, which will be launched on board the geostationary MTG-S in 2024. Once the instrument is operational, it will provide data of the full Earth disk with a 4 km spatial sampling at nadir.

The IRS scanning sequence will divide the Earth disk into four Local Area Coverage (LAC) zones, as in Figure 1. Each LAC will be covered in 15 minutes through the acquisition of successive stares, called "dwells", in about 10 seconds each. Each

---

[1]https://user.eumetsat.int/resources/user-guides/mtg-irs-level-1-data-guide, last access: 02 Jan 2024

[2]Acronym from French *Modèle de Chimie Atmosphérique de Grande Echelle*





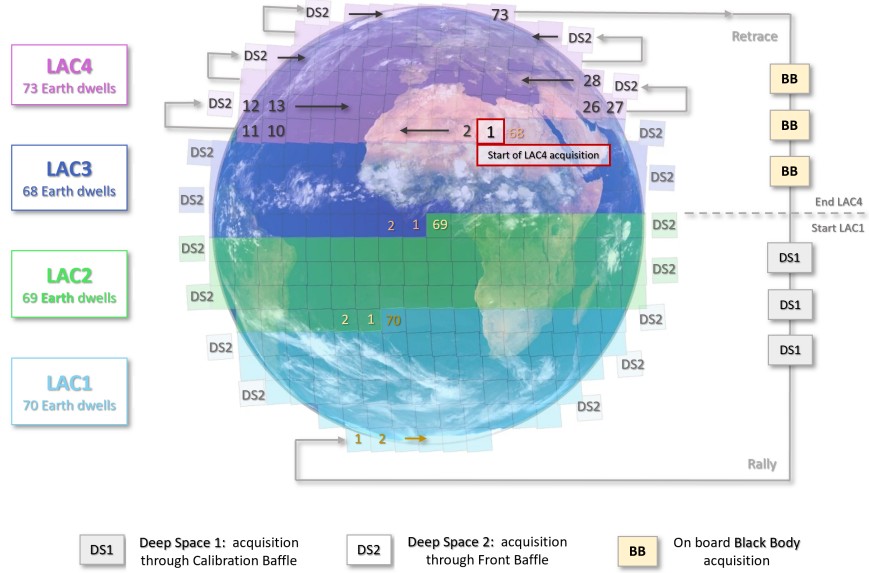

**Figure 1.** IRS Local Area Coverage (LAC) zones and dwells coverage. The geometry of acquisition is suggested starting from the first dwell in LAC1. Figure inspired by EUMETSAT portal and redesigned.

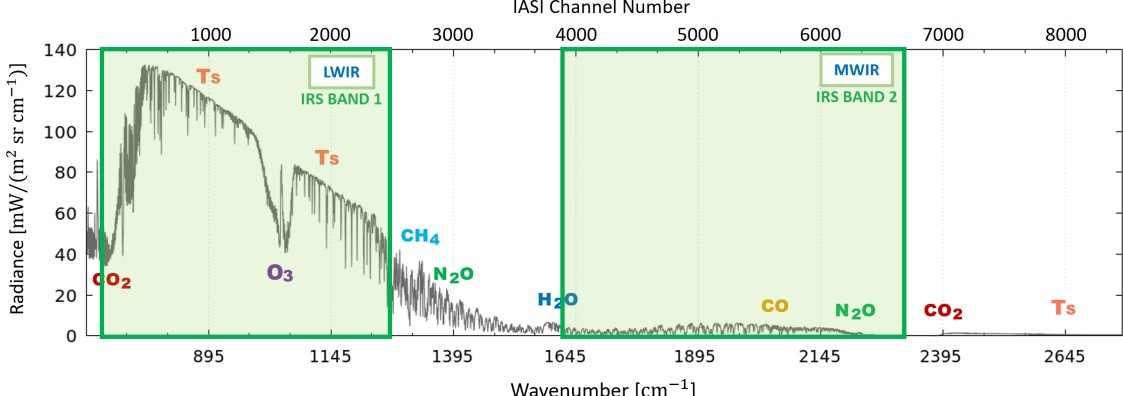

**Figure 2.** The two bands that IRS will cover are here highlighted in green on the infrared spectrum covered by IASI (in radiance units). Band 1 in LWIR goes from 679.70 to 1210.44 $cm^{-1}$, while band 2 in the MWIR is bounded in $1600.00 - 2250.20\,cm^{-1}$. Sensitivities to different species are also highlighted (Ts means surface temperature).

dwell will consist of $160 \times 160$ pixels. The total amount of Earth dwells for the whole disk will be 280, while the number per LAC is listed in Figure 1. LAC4 will be covered every 30 minutes, while the other LACs will be acquired in between.





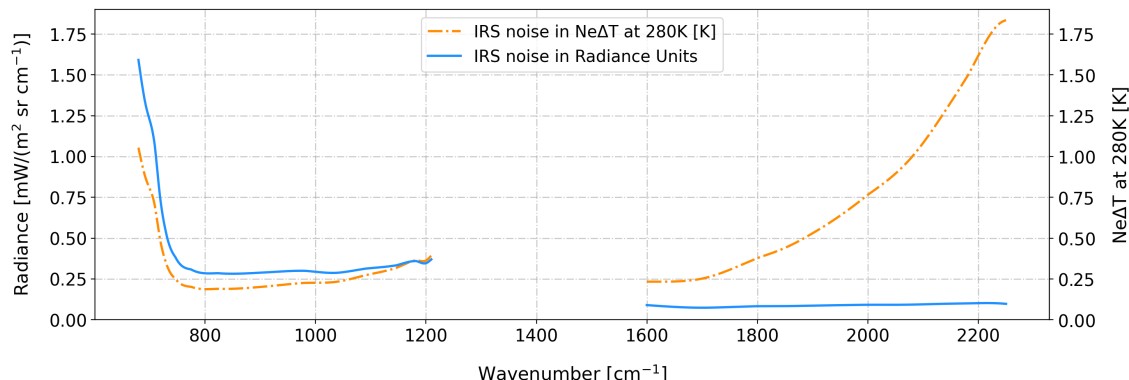

**Figure 3.** IRS instrumental noise in NE$\Delta$T at 280 K (dash-dot line) and radiance units (solid line).

The sounder will cover 1960 channels spread on two bands in the thermal infrared. In order to identify the spectral location of the two bands, they have been highlighted (in green) on a IASI spectrum in radiance units. *Band 1* in the LWIR will be bounded in the range $679.70 - 1210.44 \, \text{cm}^{-1}$, with a spectral sampling of $\sim 0.603 \, \text{cm}^{-1}$, for a total amount of 881 channels). *Band 2*, in the MWIR, in the range $1600.00 - 2250.20 \, \text{cm}^{-1}$ and with spectral sampling of $\sim 0.604 \, \text{cm}^{-1}$, will cover 1079 channels. The difference in the spectral sampling is due to the on-ground maximum optical depth ($\delta_{max}$), which is slightly different for the two bands: $0.829038 \, \text{cm}^{-1}$ for Band 1 (LWIR) and $0.828245 \, \text{cm}^{-1}$ for Band 2 (MWIR). The central wavenumber $w_n$ is computed as follows:

$$w_n = \frac{\text{N}}{2\delta_{max}} \tag{1}$$

with $1127 \leq \text{N} \leq 2007$ for the LWIR and $2650 \leq \text{N} \leq 3728$ in the MWIR [from personal communication with B. Theodore (EUMETSAT)].

The instrument noise, provided by EUMETSAT in terms of Noise Equivalent of Differential Temperature (NE$\Delta$T) at 280 K, is depicted in Figure 3 (dash-dot line). This noise can be converted to the corresponding scene temperature by using the formula:

$$\text{NE}\Delta\text{T}(\text{T}_\text{b}) = \text{NE}\Delta\text{T}(280 \, K) \frac{(\partial B_\nu / \partial T)(280 \, K)}{(\partial B_\nu / \partial T)(\text{T}_\text{b})} \tag{2}$$

Values of the noise in Radiance Units are in shown in Figure 3 using a solid line.

IRS spectra will be distributed in the form of Principal Component (PC) scores. Its 1960 channel acquisitions will be compressed in to 300 PC scores preserving information content for near-real time applications. As this study aims to carry out a preliminary analysis, however, this step will be not mimicked.





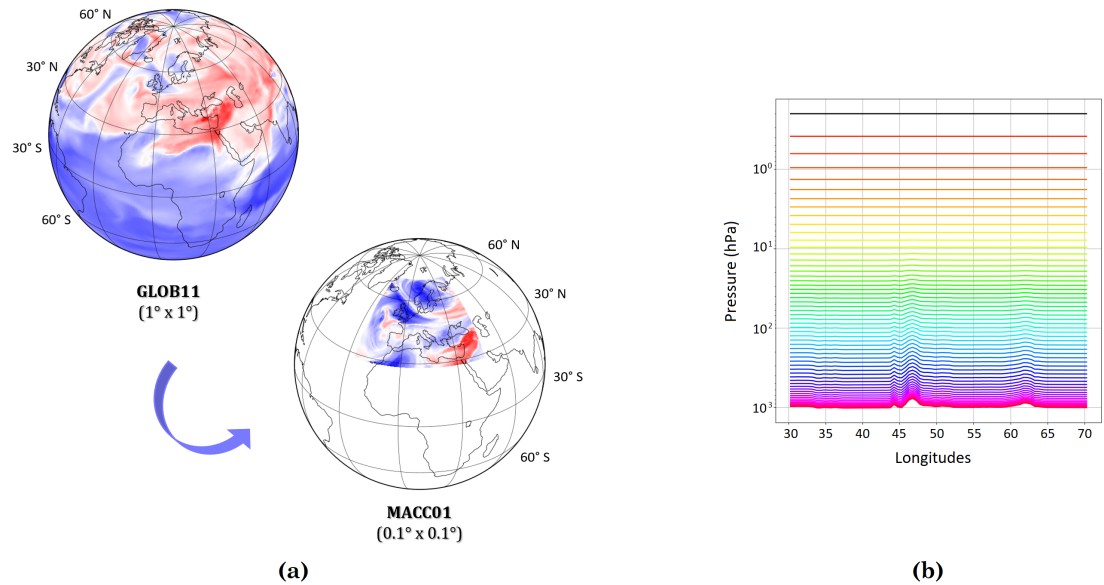

**Figure 4. (a)** Illustration of MOCAGE global domain GLOB11, which presents a $1°$lat $\times 1°$lon horizontal resolution, and the regional domain over Europe, named MACC01, with a thinner resolution of $0.1°$lat $\times 0.1°$lon, and bounded in $28°$N, $26°$W and $72°$N, $46°$E. Colors indicate ozone field at a given level. **(b)** [adapted from El Aabaribaoune (2022)] MOCAGE vertical configuration of the 60 hybrid $\sigma$-pressure levels (each one represented with a different colour).

## 3   Chemistry Transport Model

The Modèle de Chimie Atmosphérique de Grande Echelle (MOCAGE) is a three-dimensional chemistry transport model
developed at the Centre National de Recherches Météorologiques (CNRM) since 2000 [e.g. Josse et al. (2004); Sič et al. (2015); Guth et al. (2016)]. It has been exploited along the two last decades for a wide range of operational and research applications. Among the others, for instance, it served for several studies aiming to evaluate the climate change impact on atmospheric chemistry [e.g. Teyssèdre et al. (2007); Lacressonnière et al. (2012); Lamarque et al. (2013); Watson et al. (2016)], as well as the trace gases transport throughout the troposphere [Morgenstern et al. (2017); Orbe et al. (2018)]. Many efforts have
been performed to employ MOCAGE to investigate the exchanges taking place between troposphere and stratosphere using data assimilation [e.g. El Amraoui et al. (2010); Barré et al. (2014)], or also to extend the representation of aerosols within the model simulations through the aerosol optical depths (AODs) assimilation [e.g. Sič et al. (2016); Descheemaecker et al. (2019); El Amraoui et al. (2022)]. The model is also a precious resource for the air quality monitoring and forecasting on the French Prev'Air platform [Rouil et al. (2009)] and over Europe within the Monitoring Atmospheric Composition and Climate
(MACC) project [Marécal et al. (2015)].

At present, MOCAGE disposes of two geographical configurations, using a two-way nested grids capacity (Figure 4.a):

- GLOB11: a global scale with a $1°$ longitude x $1°$ latitude horizontal resolution;



- MACC01: a regional domain, bounded in 28°N, 26°W and 72°N, 46°E, and with a resolution of 0.1° longitude x 0.1° latitude (approximately 10 km at the latitude of 45°N) centered over Europe.

For the vertical levels, MOCAGE uses $\sigma$-pressure vertical coordinates Eckermann (2009). Through this system, on the calculation of which we will not enter into detail in this paper, each grid point is represented by a value above the surface and the topography is not a constraint. The model hence has a non-uniform vertical resolution: 47 vertical altitude-pressure levels from the surface up to 5 hPa. The levels are denser near the surface, with a resolution of about 40 m in the lower troposphere and 800 m in the lower stratosphere. A 60 hybrid levels version is also used in research mode and it is also the one that has

been exploited for the present work. This consists of the 47 levels computed as just described, plus 13 additional levels going up to 0.1 hPa. Resolutions in upper stratosphere is around 2 km. Please note that the MOCAGE vertical levels are numbered in descending order from the ground: the level the closest to the ground is the number 60, while the highest in the atmosphere is level 1. A schematic representation of the MOCAGE vertical configuration of the 60-hybrid pressure level is provided in Figure 4.b, while an equivalence among vertical level number and pressure can be found in Appendix A.

The model is able to simulate the chemistry in the lower stratosphere and troposphere, taking into account in detail photochemical processes and the transport of longer-lived species. It uses different chemical schemes in order to reproduce the atmospheric chemical composition: the REactive Processes Ruling the Ozone BUdget in the Stratosphere (REPROBUS) is used for the stratosphere [Lefevre et al. (1994)], while for the tropospheric representation the Regional Atmospheric Chemistry Mechanism (RACM) is exploited [Stockwell et al. (1997)]. Through the combined use of the two aforementioned schemes

(called RACMOBUS), MOCAGE is able to simulate 118 gaseous species, 434 chemical reactions, primary aerosols and secondary inorganic aerosols.

MOCAGE CTM runs in an off-line mode. Depending on the application it can be coupled with a general circulation climate model (for climate studies), or with NWP models (e.g. for near real time applications). The core of the chemical reactions used in MOCAGE is also exploited on-line into the Integrated Forecast System (IFS) Huijnen et al. (2019); Williams et al. (2022). In

this study MOCAGE has been used off-line and the meteorological forcing comes from the Météo-France's operational global NWP model ARPEGE (Action de Recherche Petite Echelle Grande Echelle) Courtier et al. (1991), in a configuration with the two domains GLOB11 and MACC01.

The data concerning chemical emissions are provided to MOCAGE as external data-sets. For the present operational configuration, MEGAN-MACC [Sindelarova et al. (2014)] and the Global Emissions Inventory Activity (GEIA) are used for biogenic

emissions. MACCity [Lamarque et al. (2010); Granier et al. (2011); Diehl et al. (2012)], RCP60 [Fujino et al. (2006); Van Vuuren et al. (2011)], CAMS-REG-AP [Guevara et al. (2022); Kuenen et al. (2022)] and GEIA are those provided to cover the information about anthropogenic emissions. For the representation of the biomass burning, data from the CAMS Global Fire Assimilation System (GFAS), from ECMWF, are exploited [Kaiser et al. (2012)].





## 4  Data Assimilation Methods

The assimilation system used within MOCAGE has been originally defined in the ASSET (Assimilation of Envisat) project
[Lahoz et al. (2007)]. It was jointly developed by CERFACS (Centre Européen de Recherche et de Formation Avancée en
Calcul Scientifique) and Météo France and further refined over the years. It has already been exploited for many studies on
the assimilation of chemical data [e.g Massart et al. (2009); Emili et al. (2014, 2019); El Aabaribaoune et al. (2021)], and also
on aerosols assimilation [e.g. Sič et al. (2015); Descheemaecker et al. (2019); El Amraoui et al. (2022)], in the study of the
exchanges between troposphere and stratosphere [e.g. El Amraoui et al. (2010)] and in many other fields.

For this work we use the three-dimensional variational (3D-Var) method with a hourly assimilation window. The aim of
the method is to look for the best representation of the atmospheric state, or in other words the best compromise between the
background model state and the observations. This is done by minimising the cost function that follows:

$$J(\mathbf{x}) = \frac{1}{2}\left(\mathbf{x} - \mathbf{x}^{b}\right)^{T} \mathbf{B}^{-1} \left(\mathbf{x} - \mathbf{x}^{b}\right) + \frac{1}{2}\left[\mathbf{y} - H(\mathbf{x})\right]^{T} \mathbf{R}^{-1} \left[\mathbf{y} - H(\mathbf{x})\right] \tag{3}$$

where $\mathbf{y}$ is the vector of the observations, while $\mathbf{x}_b$ and $\mathbf{x}$ the *a priori* background and the model state vector respectively.
The state that minimises the cost function $J(\mathbf{x})$ will then be defined as $\mathbf{x}_a$, i.e. the analysis state. $H$ is the observation operator,
that is usually a Radiative Transfer Model (RTM) whose function is to transform a model state to a vector comparable to the
observed radiances (or *vice versa*). In this work this function is covered by the Radiative Transfer for TOVS (RTTOV) version
12 [Saunders et al. (2018)] in clear-sky conditions (the scattering effect of clouds and aerosols are not taken into account).
Last but not least, the covariance matrices of the background and observation errors (**B** and **R** respectively) are two essential
components in the equation, since they allow each term to be given its proper weight.

### 4.1  Observation Error Estimation

In the past years the importance of representing off-diagonal correlations has emerged, especially for satellite data. As a result
of the above, it often proves necessary to structure **R** matrices containing non-zero covariance terms in order to take into
account the inter-channel correlations.

Throughout this work, a procedure introduced by Desroziers et al. (2005), and for this reason usually referred to as "Desroziers's
diagnostics", is the one used to estimate the structure of a full **R** matrix. This technique is very efficient and, in addition to
having been used over the years for a wide range of researches [e.g. Weston et al. (2014); El Aabaribaoune et al. (2021);
Vittorioso et al. (2021)], it is currently exploited in the operations by many meteorological centres.
Through this method variances and covariances of observation errors can be obtained from innovations $\mathbf{d}_{b}^{o} = \left[\mathbf{y} - H(\mathbf{x}_{b})\right]$
and residuals $\mathbf{d}_{a}^{o} = \left[\mathbf{y} - H(\mathbf{x}_{a})\right]$ statistics. This matrix is then given by the expression:

$$\mathbf{R} = E\left\{\left[\mathbf{y} - H(\mathbf{x}_{a})\right]\left[\mathbf{y} - H(\mathbf{x}_{b})\right]^{T}\right\} \tag{4}$$

where $E$ is the statistical expectation operator.



## 4.2 Background Error Estimation

In previous studies the background-error standard deviation was assumed to be proportional to the ozone concentration itself. Emili et al. (2014) and Peiro et al. (2018) chose to use an error varying along the vertical column and expressed as a percentage of the $O_3$ background profile: a percentage of $15\%$ was attributed to the troposphere and a smaller one of $5\%$ to the stratosphere. Emili et al. (2019), comparing the standard deviation of a free model simulation against independent observations, actually, show that the error is lower in stratosphere, larger in free troposphere with the highest values near the tropopause. In this latter

study and in the follow-up El Aabaribaoune et al. (2021), however, these percentages have been refined up to the value of $2\%$ above 50 hPa and $10\%$ below, since the model itself had been upgraded compared to the prior works.

   In the present study, as a more recent version of the model MOCAGE is used, we prescribe $2\%$ all over the entire atmospheric column in order to compute the background standard deviation, i.e. the square root of the diagonal of the first $\mathbf{B}$ we evaluate. In other words, $2\%$ of the ozone concentration at each atmospheric level was attributed to the background standard deviation

$(\sigma_B)$. The variances (i.e. $\sigma_B^2$) were then computed and attributed to the diagonal. The correlation terms have been modelled using a diffusion operator [as in Emili et al. (2019) and El Aabaribaoune et al. (2021)].

## 5   OSSE

   In the context of this work, it is necessary to have good quality and reliable simulated spectral radiances in order to assess the impact of IRS assimilation into the CTM MOCAGE. Since IRS is not yet flying at the time the research is carried out, the

strategy we adopted, and the one most commonly adopted in the atmospheric sciences in similar scenarios, is to perform an Observing System Simulation Experiment (OSSE) [among others Errico et al. (2007); Masutani et al. (2010); Claeyman et al. (2011); McCarty et al. (2012); Privé et al. (2013a); Privé et al. (2013b); Boukabara et al. (2016); Duruisseau et al. (2017); Descheemaecker et al. (2019); Zeng et al. (2020); Coopmann et al. (2023)]. This kind of experiments comply a series of steps and validations to ensure that the observations are reliable, truthful and to put them to test into a data-assimilation system.

An OSSE basically consists in simulating synthetic observations from an atmospheric model state representing reality, assimilating them within another model state representing the model itself, and finally evaluating their impact on analyses and forecasts.

   The reference reality is usually referred to as Nature Run (NR). This consists of an atmospheric state that must, in fact, realistically reproduce the true state of the atmosphere. It will serve, throughout the experiment, as the state from which

observations will be simulated and as the reference against which the final assimilations will be verified. It is usually produced using a good quality model in free-run, or not providing any information coming from real observations.

   The NR reality is used to feed an observation simulator, that is usually an RTM through which the sought synthetic observations will be produced. Into the simulator some specific instrumental proprieties have to be specified (such us optics, observational geometry, spatial and temporal resolution). The perfect observations obtained are then perturbed with an instru-

mental error, to be properly assessed, in order to reach their final shape.



Another fundamental step in the development of an OSSE is the creation of the so-called Control Run (CR), that is a run of a model simulating the reality. The differences between the NR and CR should be those existing between the reality itself and the output of a good quality model trying to reproduce it.

The synthetic observations will then be assimilated into the CR. This final run, referred to as Assimilation Run (AR), is realized with the same configuration as the CR, by assimilating the synthetic observations created from the NR.

Finally, the impact of assimilation is assessed by comparing the results of the assimilation to the reference reality (i.e. the NR) and to the run without assimilation (i.e. the CR).

For an OSSE to be considered robust, it is very important that the CR is consistent with the NR, but different enough to avoid the so-called "identical twin" problem [e.g. Arnold Jr and Dey (1986); Masutani et al. (2010)]. This can be avoided by using different models to create the two scenarios, or by sufficiently differentiating the inputs and the configurations in the same model, so that the errors are properly represented and the outputs are consistent but divergent at the same time. In any case, the spatio-temporal variability in NR must be properly evaluated against the CR before assimilation is carried out [Timmermans et al. (2015)].

### 5.1 OSSE framework

The diagram in Figure 5 visually summarises the strategy adopted for the creation of the OSSE specific for this study. First of all, MOCAGE was the model chosen for the creation of both CR and NR. In both cases, both global and regional domain were activated and the meteorological forcing came from the model Action de Recherche Petite Echelle Grande Echelle (ARPEGE) in its version operational in Météo-France in 2020. Consequently, since the settings exposed so far are identical, a big effort had to be done in order to plan a strategy avoiding the identical twin problem. The first step was to differentiate the surface emissions to be used for the two runs. For biogenic and anthropogenic emissions in the NR framework, the configurations used in operational MOCAGE at the time this work begun, have been employed for each geographical domain. For the CR framework, on the other hand, data referring to the year 2000 were used. This provided the same kind of spatial variability for this class of emissions than the NR, but with different intensities. For the representation of the biomass burning, data from the CAMS GFAS, were used as input to MOCAGE for the NR settings, while for the CR was MACCity representative of the year 2000. For the details about the emissions provided to MOCAGE for each run and domain, see Table 1.

In addition to the modulation of the surface emissions, another action was taken in order to have a CR and a NR consistent with each other, but different enough to reproduce the differences that would exist between the true reality and a model that reproduces it. Radiances from the Metop-IASI spectrometer have been assimilated into the MOCAGE model in the GLOB11 configuration. However, the regional domain MACC01 nested in the global, and in which no assimilation was directly carried out, has been the NR employed for the simulation of IRS observations.

A MOCAGE run, which has been named "pure", was equally planned and performed. It exploited the same meteorological input as the other runs. No assimilation was performed in it, like the CR. At the same time, it the same surface emissions as the NR were provided. As a consequence, such a run provides a reference in order to evaluate the impact of the IASI radiance assimilation in the NR and, at the same time, the contribution of the surface emission modulation in the CR.





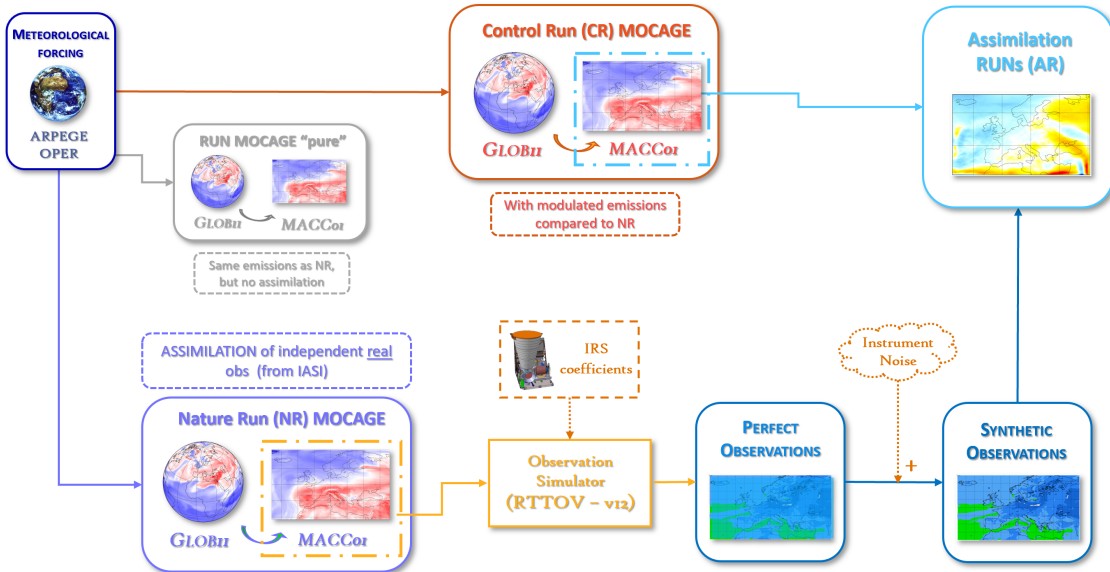

**Figure 5.** Implementation of an OSSE specific to the present work.

Next, the simulation of IRS radiances was carried out from the NR on the MACC01 domain, through the RTTOV v12 RTM. "Perfect", i.e. noise-free, observations were created (details in the following).

    The observations produced over MACC01, for both instrumental configurations, were then assimilated into the MACC01 domain of the CR.

    The time period running from the 1st of May to the 31st of August 2019 has been chosen to carry out the experiments.
More specifically, the runs without assimilation, i.e. CR and NR, started on the day May the 1st. Assimilation into the global IASI L1c radiances began on May 15th, which left 15 days of spin-up time for the system. An additional 10 days, were left as spin-up before debuting the assimilation of IRS radiances into the CR. The ARs, thus began from May 25th. Each evaluation on the results, however, was made starting from the 1st of June.

### 5.2   Control Run vs Nature Run

Once the Nature Run that best suited the purposes of this research was obtained, an evaluation against the CR was a mandatory step. This is particularly important for the present study since the same model was used for both the runs and an inter-comparison was needed to verify that the precautions taken to differentiate the configurations were sufficient.

    At first, $O_x$ concentrations in total column are averaged over the 3-month period of study, from June to August, for both the NR and the CR. The variations between these averages are assessed by computing relative difference on percentage as follows:
$D_{rel\%} = [(\overline{CR} - \overline{NR})/(\overline{NR})] \cdot 100$ where the NR is used as the reference. By reviewing the results, displayed in Figure 6.a, one can estimate that the variations are always negative. In other words, the CR shows concentrations of $O_x$ that are always



| | CR framework | | NR framework | |
|---|---|---|---|---|
| | GLOB11 | MACC01 | GLOB11 | MACC01 |
| Meteorological forcings | ARPEGE OPER | ARPEGE OPER | ARPEGE OPER | ARPEGE OPER |
| Biogenic emissions | CAMS-GLOB-BIO year 2000 Granier et al. (2019) Sindelarova et al. (2014) CAMS-GLOB-SOIL for NOx (year 2000) Granier et al. (2019) Simpson and Darras (2021) | CAMS-GLOB-BIO (year 2000) CAMS-GLOBE-SOIL for NOx (year 2000) | MEGAN-MACC (year 2010) Sindelarova et al. (2014) GEIA for NOx (year 1990) | MEGAN-MACC (year 2010) GEIA for NOx (year 1990) |
| Anthropic emissions | CAMS-GLOB-ANT (year 2000) Granier et al. (2019) Kuenen et al. (2022) Global Emission Inventory Activity (for chlorine species) (GEIA) | CAMS-REG-AP (year 2000) Guevara et al. (2022) GEIA (for chlorine species) | MACCity (year 2016) RCP60 (year 2016) Fujino et al. (2006) Van Vuuren et al. (2011) | CAMS-REG-AP (year 2017) GEIA (for chlorine species) |
| Biomass burning | MACCity (year 2000) Lamarque et al. (2010) Granier et al. (2011); Diehl et al. (2012) | MACCity (year 2000) | Global Fire Assimilation System (GFAS) (year 2019) Kaiser et al. (2012) | GFAS (year 2019) |
| Data assimilation | None | None | IASI L1C | None |

**Table 1.** Summary of the different settings chosen for the CR and the NR frameworks. The boxes in green indicate that the same parameters have been employed for both runs, given the same MOCAGE geographical domain. The boxes in red shades, on the other hand, highlight the parameters that have been differentiated.

lower than for the NR. More specifically, the deviation between the average concentration ranges between a minimum of 10.5% and maximum of 13.5%.

The standard deviation of the difference is also evaluated and shown in Figure 6.b. The main strongest values are encountered
over the continental Europe (between 20 and 35°E, 50 and 60°N) up to almost 13 DU. The weakest error values are instead found at the south-eastern edge of the domain.

Typical ozone concentrations in the troposphere are lower than those found in stratosphere. When performing a study on the total columns, thus, the contribution of stratospheric ozone will be the one that mainly arises. In order to better assess what happens in troposphere, then, a study limited to the tropospheric layer must be led. After empirically assessing the average
position of the tropopause, and cross-comparing it with the vertical levels provided by MOCAGE, it has been decided to

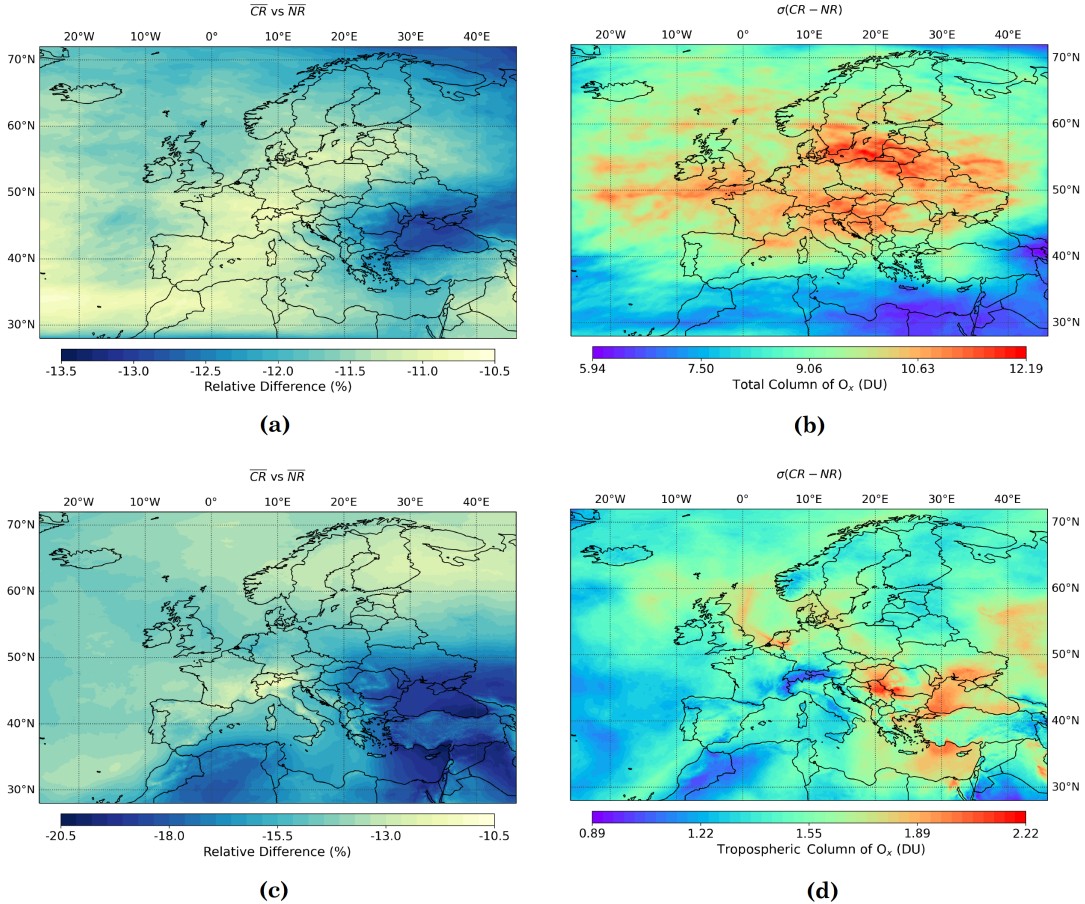

**Figure 6.** Relative difference between the averages over the 3-month period of study (1st of June up to 31st of August, 2019) of the $O_x$ Total Columns from the CR and the NR **(a)**. Panel **(b)** is the Standard Deviation of their differences. The corresponding values for Tropospheric Columns are shown in **(c)** and **(d)**.

define in this manuscript as *Tropospheric Column* the one running from the surface up to about $300\,hPa$, which corresponds to MOCAGE vertical levels ranging from 60 to 40 (see Appendix A).

In comparison to what had been observed for the total columns, when quantifying the differences occurring between the two cases as in Panel 6.c, it is observed that these reach percentage values that are even higher than in the case of the total columns. The maximum variations occur in the South-East quadrant of the domain, with peaks of -20% (CR values smaller than NR) above the Black Sea. The standard deviation of the bias between the two scenarios, shown in Panel 6.d, reports areas of minima above Alps, Morocco and in the Middle-East area. Maxima, on the other hand, are over Netherlands and the nearby seaside area, above the western portion of the Black Sea, southward on the Mediterranean and Egypt, in the continental area





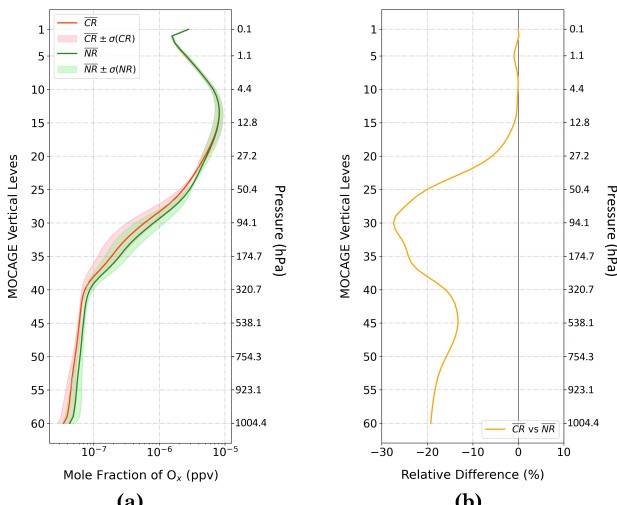

**Figure 7.** Average over the 3-month period of study (1st of June up to 31st of August, 2019) and, for each of the 60 MOCAGE model levels, on the MACC01 domain for both CR and NR, plus and minus their standard deviations **(a)**; relative difference between the averages is also shown in **(b)**.

around 45°N and 26°E. Some of these values may be led by stratospheric intrusions, which are known to take place in the Mediterranean basin [Lelieveld and Dentener (2000); Lelieveld et al. (2002)].

Another inter-comparison between CR and NR was also performed by averaging, not only over the 3-month period of study, but all over the regional domain. The averages of the $O_x$ concentrations obtained for each of the 60 MOCAGE levels, plus and minus the respective standard deviations, are shown in Figure 7 (a). The relative difference is also evaluated between the two runs in percentage terms and are displayed in Panel 7.b. The two scenarios seem to mostly diverge in the lower troposphere and in stratosphere between levels from 40 to 25, i.e. approximately between 320 at 50 hPa. More in detail, variations of the order of 20% for averages occur in the lower layers. The maxima, on the other hand, are found around 90 hPa), where the NR averaged values appear to be stronger by 28% than the CR ones. On the other hand, NR and CR are very comparable for levels 20 and above. The results of the inter-comparisons just shown are considered enough to avoid the "identical twin" problem.

## 5.3 Simulated Observations

IRS characteristics can be essential in deducing information about atmospheric composition. Looking at the spectral bands that the instrument will cover (e.g. Figure 2), it is clear that IRS will be sensitive to different chemical species. Despite this *a priori* knowledge about the potential of the instrument, however, a prior sub-band selection of a spectral sub-set of wavelengths to work on was necessary. As IRS is not yet operational the research will require simulation of data using models, either RTM or CTM. Each simulation presents a certain computational cost and it is time-consuming. Therefore, for this work the IRS sub-band containing 195 contiguous channels between 982.464 and 1099.467 cm$^{-1}$ [i.e IRS channel number 503 - 697] has been



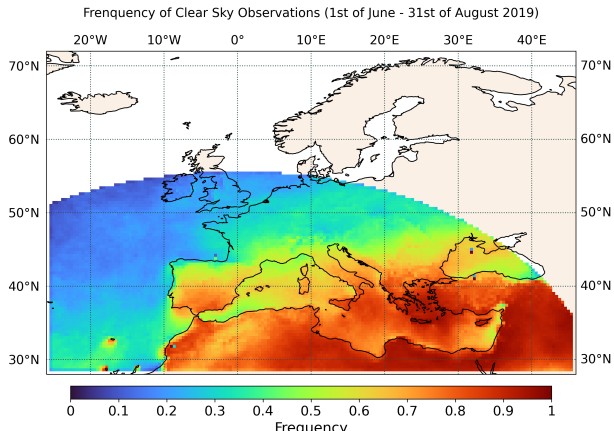

**Figure 8.** Frequency of the simulated clear-sky observations over the 3-month period of study (from June to August 2019).

retained for the simulation of the observations. This band is centred on the portion of the spectrum with the highest sensitivity to the ozone species. However, this also includes, at the edges, wavelengths in the atmospheric window, and with a mixed sensitivity. This is an element to take into consideration in the evaluation and conclusion sections. The choice of this band was consolidated by performing sensitivity studies (not shown) exploiting a set of profiles gathered from the diverse profile data

sets on the Copernicus Atmosphere Monitoring Service (CAMS) atmospheric composition forecasting system, provided by the NWP SAF [3] , and applying the so-called *physical selection method*, suggested by Gambacorta and Barnet (2012).

Each simulation was performed through RTTOV version 12 with radiative transfer coefficients provided by the RTTOV team. Specific coefficients for IRS have been built by the RTTOV team and are those exploited throughout this work.

Since, at present, only clear-sky observations are assimilated in MOCAGE, only clear-sky pixels are simulated for this

work to save computation resources. To determine whether a pixel is clear or not, the model cloud parameters are used, which come, therefore, from the meteorological forcing exploited (i.e. ARPEGE OPER). Given the density of the observations that an instrument of the IRS's calibre will be able to provide, it becomes unlikely, from computational point of view, to be able to simulate and then assimilate such dense observations in the time available for this project. Plus, we are not able to assimilate a dense observation network as we are not yet able to take into account horizontal correlation between observation

points. A thinning of the pixels to be simulated was therefore carried out. One pixel per $0.4°$ box was therefore simulated in each scenario. To the perfect observations thus obtained, the IRS instrumental noise was then added to produce the ultimate synthetic observations.

Figure 8 report the frequency of the simulated clear-sky observations all over the 3-month period of study. It is evident that observations are more dense over land and in the South-East portion of the instrument disk, most likely due to a sparse cloud

cover over these areas during these summer months. The lowest density, on the other hand, occurs over the Atlantic Ocean,

---

[3]https://nwp-saf.eumetsat.int/site/software/atmospheric-profile-data/





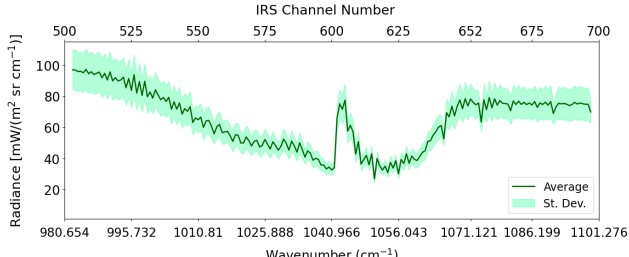

**Figure 9.** Simulated noised spectrum for one day in the period of study (1st of July, 2019), averaged over the hours of the day, plus and minus standard deviation.

reasonably for the opposite reason. This kind of illustration also provides a view of the area covered by the simulated IRS observations.

Figure 9 shows the simulated, and noised, IRS ozone spectral sub-band (in radiance units) as an average over one day along the whole simulated period (1st of July, 2019). The average is computed over the 24 hours, together with the standard deviation. The characteristic signature of ozone in the spectrum is well observed and standard deviation of the simulations remains more or less constant over the band.

## 6   Results

### 6.1   Assimilation Set-Up

The assimilation of the synthetic observations was carried out inside the CR (on MACC01 domain) in the time period going from the 15th of May, till the 31st of August, 2019. The evaluation was then performed in the three months of June, July and August. As already done during the work for the preparation of the NR, the assimilation algorithm used was the 3D-Var with a hourly assimilation window. The role of the observation operator $H$ was covered, once again, by the RTTOV version 12 in clear-sky conditions (the scattering by aerosols and clouds was not taken into account).

The observation error was computed through the Desroziers's method previously illustrated. As already explained, this kind of procedure is used to compute full **R** matrices, which has non-zero covariance terms, using observations, background and analysis. In order to have an initial analysis to use for this purpose, a first assimilation was performed using a diagonal **R** matrix. The variance values forming the diagonal have been determined using a fixed standard deviation $\sigma = 2.0\,\mathrm{mW/(m^2\,sr\,cm^{-1})}$. This value was chosen so as to exceed the average values of the instrumental noise.

Once the first analysis was available, the full **R** was computed through Equation 4 .

The diagnosed standard deviation is shown in Panel (a) of Figure 10 together with the corresponding instrumental noise. The diagnosed standard deviation shows higher values and most significant spread from the instrument noise, between around $1005\,\mathrm{cm}{-}1$ and $1060\,\mathrm{cm}{-}1$. This part is the most sensitive to zone. The small error in the observation operator in the ozone



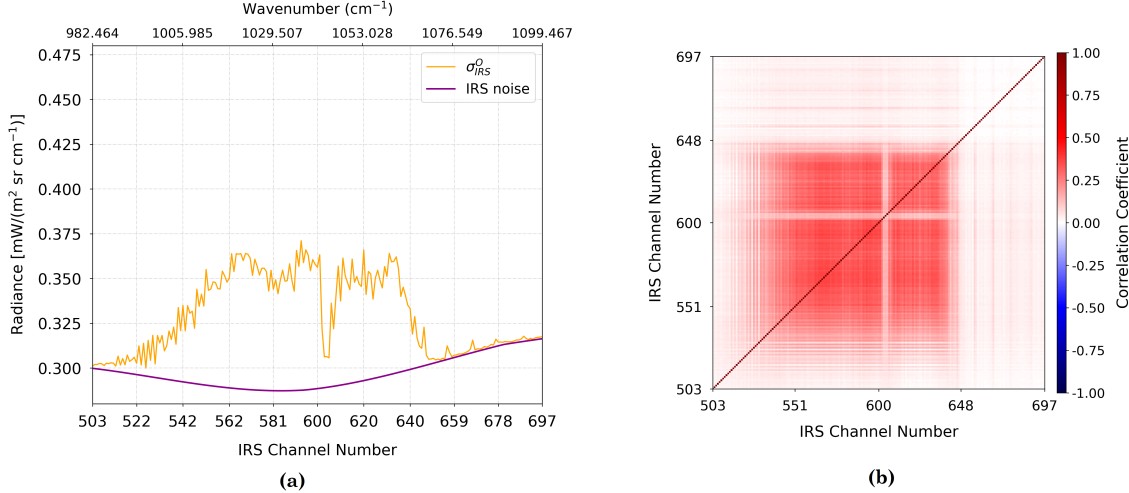

**Figure 10.** Diagnosed observation-error standard deviations ($\sigma_{\text{IRS}}^{\text{O}}$) and instrument noise for IRS in **(a)** and diagnosed error correlations in **(b)**.

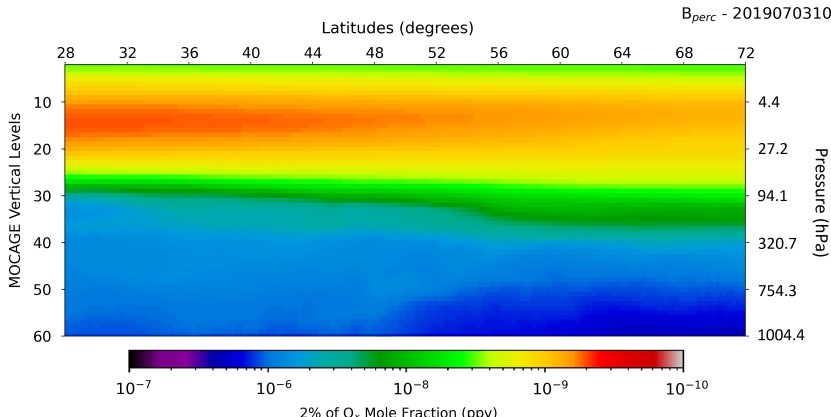

**Figure 11.** Example of **B** matrix, obtained as $2\%$ of the ozone concentration, for one random day and hour inside the period of study (3rd of July, 2019, 10 UTC). Values are averaged over longitudes.

treatment can explain this additional contribution to the whole observation error. The diagnosed correlations are displayed in Figure 10.b. The most correlated portions are also to be related to the observation operator error covariance.

The **B** matrix was obtained as introduced in Subsection 4.2. An example of the shape of $\mathbf{B}_{\text{perc}}$ is provided in Figure 11 for one day and hour (3rd of July, 2019, 10 UTC) representative of the general behaviour of the period of study, over the MACC01 domain. The strongest values are found in stratosphere, between levels 20 and 10 (about 27 to 4 hPa) at latitudes between 28 and 44°. The smallest values, instead, are found in the lower-most troposphere at latitudes above 48°.





## 6.2 Statistics on the Observations

The assimilation trials described in this study consist in a continuous hourly assimilation cycle over the period of study. This means that each assimilation time creates an analysis, influenced by the observations, which is the initial state of a 1-hour forecast, that is, in turn, the background state of the next assimilation time. Effects of the observation are propagated from one assimilation to the next one, reaching some steady regime in the assimilation cycle.

A first assessment of the assimilation of IRS radiances into the MOCAGE model has been carried out through the evaluation
of *Observations minus Background* (O-B), i.e. innovations, and the *Observations minus Analysis* (O-A), i.e. residuals.

Averages, and associated standard deviations, have been computed per hour of the day for each day in the 3-month time period (Figures 12). The results refer to an arbitrarily-selected wavelength among those simulated, i.e. IRS channel number 552 (1012.016 cm$^{-1}$), which is representative of most of ozone-sensitive channels in the spectral range used in this study. Such a channel presents, indeed, a Jacobian picking around ∼320 hPa), while, when weighted with the 10% of the ozone
profile, it shifts its sensitivity between 25 and 50 hPa. The averages (Panel 12 a) show residuals always smaller compared to the innovations. This is an indication of successful assimilation that produces analyses closer to the observations than the background state. The standard deviation of residuals (Panel 12 b), on the other hand, always presents smaller values than that of the innovations. This is an indication of error reduction through assimilation. In addition, values vary with respect to the hour of the day and during the study period.

## 6.3 Evaluation of the Assimilation

A verification of the analysis against the NR was then performed in order to evaluate the impact of the IRS assimilation on the $O_x$ field produced by MOCAGE.

Average and standard deviation of the differences between the two runs for total column of ozone are shown in Figure 13.a and 13.b. From an analysis of the averages, it is found that the AR is very close to the NR in the centre of the area in which
the observations are assimilated (Figure 8). The variation, more specifically, is close to 0% over the Mediterranean basin and increases progressively while approaching the edges of the assimilation area, but not exceeding -3% (i.e. NR provides slightly stronger values of Ozone total columns than the AR). Maxima of divergence of these runs are found outside the area where the observations are present (up to -13%). At the lower edge of the domain, further variations are found and could be explained by the lateral boundary conditions bringing information from outside the domain, where no IRS observations are assimilated. As
already explained, due to the continuous assimilation cycle, the background in the inner part of the domain is more consistent with observations. Conversely, at the edges the ozone field from the coupling model (global) shows more discrepancies with observations. This trend should be taken into account in the evaluation of statistics carried out on the entire regional domain. At a later stage, one may consider performing such evaluations on a smaller domain that excludes these adjustment values.

Looking at the standard deviations of the differences (Panel b of Figure 13), values on the order of a few DU are found in the
section of the domain where the IRS observations were simulated and then assimilated. Outside this area, the values rise up, reaching maxima in the North of the domain. The impact of assimilation is therefore evident, especially if comparing what was



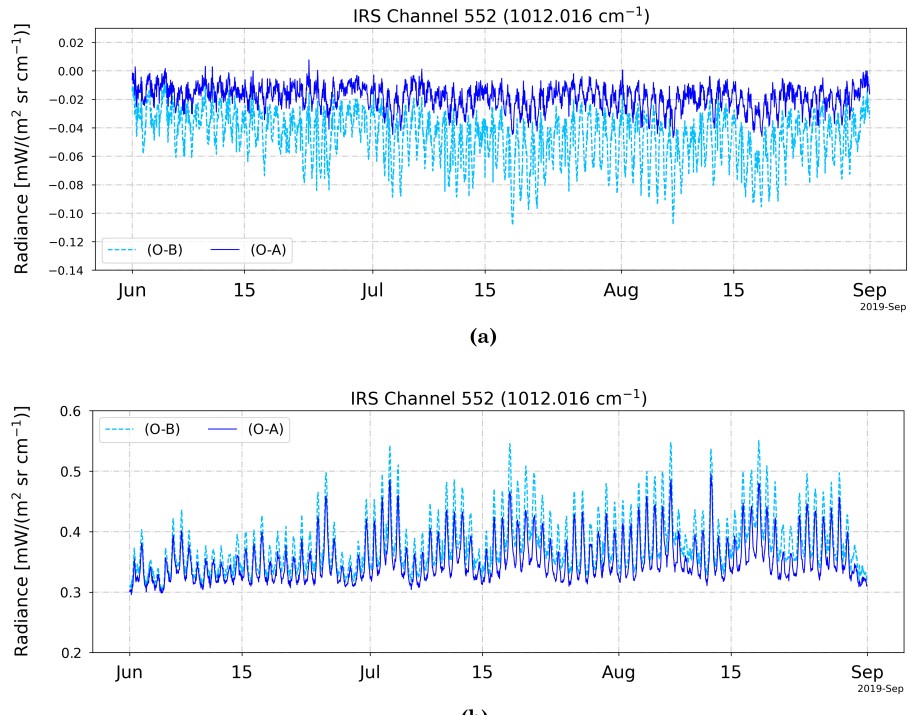

**(a)**

**(b)**

**Figure 12.** Statistics of the innovations (O-B, dashed line) and residuals (O-A, solid line) computed per each day and hour of the period of study (1st of June - 31st of August, 2019). The averages are shown in **(a)**, while standard deviations are in **(b)**. Results refer to IRS channel 552 $(1012.016\,\mathrm{cm}^{-1})$.

achieved when no assimilation was not carried out, i.e. with the CR. The analysis made in the previous Section when comparing CR and NR, in fact, found way stronger values of variation, going from -10% up to -13% for the averages. Remarkable is also the reduction in the error of the bias with respect to the NR, which, in the area where IRS radiances are assimilated, reaches
its lowest values (magenta area). The minima, around 1.82 DU, are located in the South-Est quarter of the domain, where the highest concentration of simulated, and assimilated, observations is found over the three months (look back at Figure 8).

As already done when comparing CR and NR, we also want to assess the impact of IRS assimilation on the tropospheric column. As before, this is considered to correspond to MOCAGE levels ranging between 40, i.e. $\sim 220$ to $330\,\mathrm{hPa}$, and 60, i.e. the surface (empirical evaluation based on the approximate position of the tropopause).
The relative difference between the averages Figure 13.c reports values further from zero than what was obtained for the total columns, where they were around 1%. In this case, variations of the order of 2 to 3% are observed. Although the influence of the coupling between global and regional domain at the lower border is still present, it is less pronounced than in the case of the total columns. The error of the biases shown in Figure 13.d is lower than the one found for total columns (Figure 13.b). Please notice that the color-map is the same, but this is set on a different range of values due to the lower concentrations of


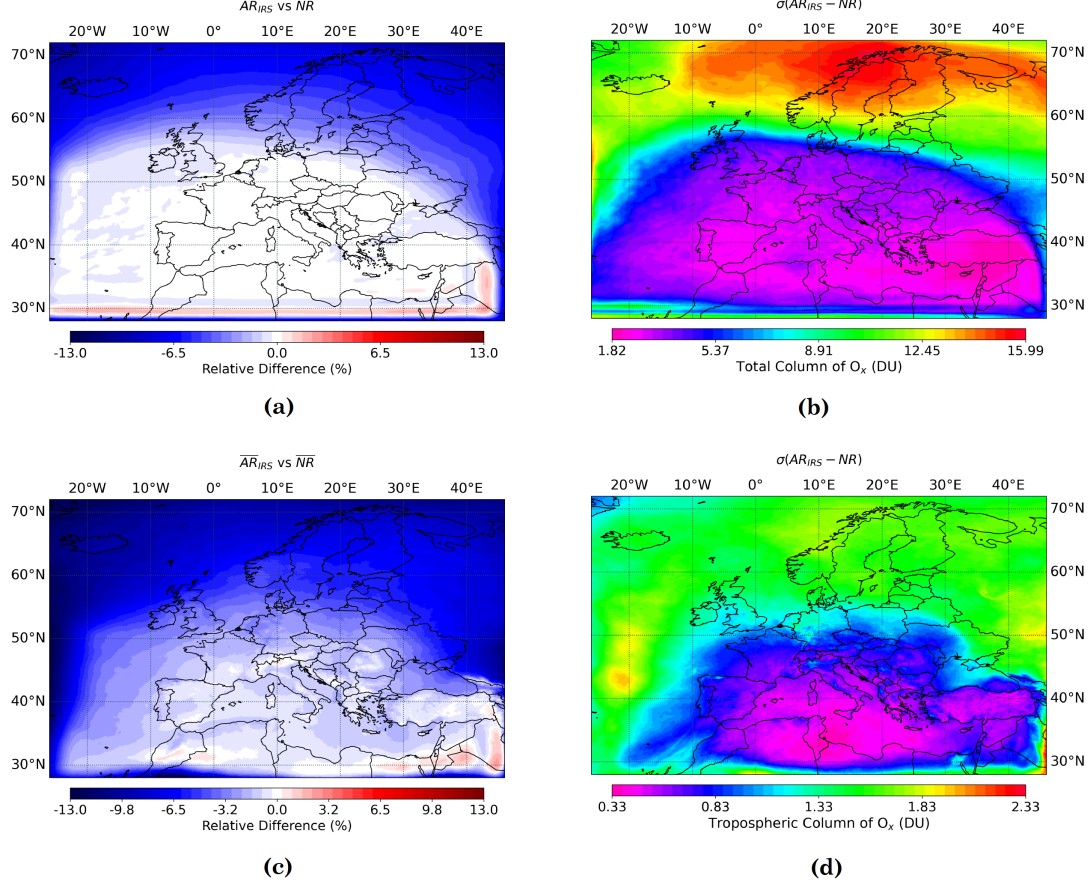

**Figure 13.** Relative difference between the averages over the 3-month period of study (1st of June up to 31st of August, 2019) of the $O_x$ Total Columns from the AR of IRS synthetic observations and the NR **(a)**. Panel **(b)** is the Standard Deviation of their differences. The corresponding values for Tropospheric Columns are shown in **(c)** and **(d)**.

the tropospheric field. In this second case, the values are bounded between 0.3 and 2.3 DU, with minima encountered on the Central-Mediterranean area and Tunisia. A different structure becomes noticeable, in this case compared to the case of the total columns, on the left side of the domain between 40 and 50°N at around 20°W, which shows slightly higher values (about $2\,DU$) of error associated with the bias between the two runs.

By averaging the ozone mole fractions over both latitudes and longitudes, further conclusions can be drawn. The normalised
average of differences between CR and NR (blue) and AR for IRS and NR (green) are depicted in Figure 14.a, while normalised standard deviations are shown Figure 14.b. The first thing one observes is that both CR and AR run being evaluated almost always provide values of $O_x$ that are weaker than NR ones. This is not the case, however, for a few model levels, like 20 or 14, where assimilating IRS simulated radiances brings to positive, even if very low, relative difference percentages. It is evident





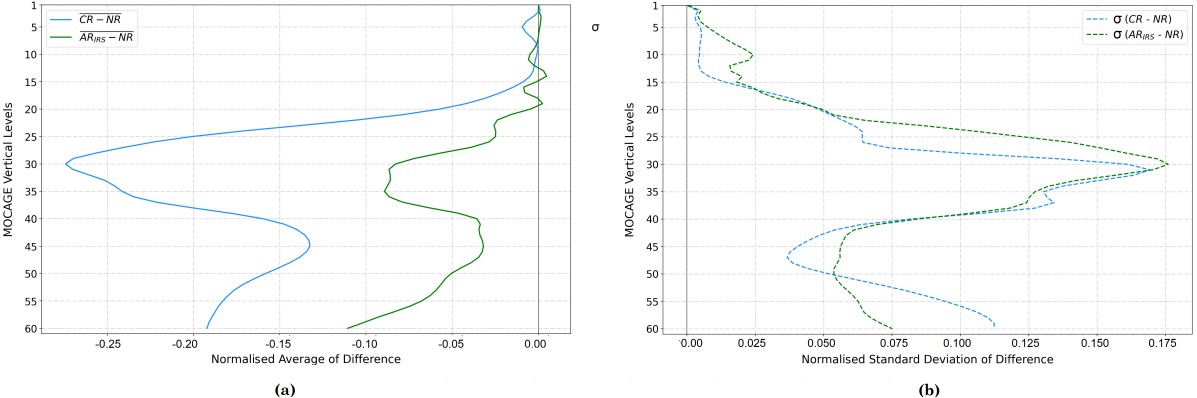

**Figure 14.** Normalised average (a) and standard deviation (b) of differences between CR and NR in blue (rp. AR for IRS and NR in green). The normalization is done with the average NR profile. The statistics are computed, for each MOCAGE vertical level, over the 3-month period of study (1st of June up to 31st of August, 2019) and the geographical MACC01 domain.

that the difference reduces when performing IRS assimilation, with a maximum improvement of $20\%$ at model level 30. At the
same level the largest error in CR is strongly reduced with the assimilation of IRS, more than $15\%$ deviation is found when no assimilation is performed compared to when it is. The comparison of the standard deviation of the differences with respect to the NR leads to less obvious impact. The standard deviation is improved when assimilating IRS in some levels (notably close to the surface), whereas it is degraded in some others. Note that the average is larger than the standard deviation, which implies that the total root mean square difference is largely decreased in the assimilation run compared to the control run. The
assimilation has an overall good impact over the model restitution, since it brings the difference compared to the NR reality closer to zero than when not performing any assimilation.

## 7 Conclusions and Perspectives

The present research took place in the very actual context of preparing the arrival of the new Infrared Sounder (IRS), that
will fly aboard the Meteosat Third Generation (MTG)-S satellites in the next few years. As already discussed in detail within
the manuscript, such an instrument will show a considerable potential for application in the sphere of atmospheric chemistry
monitoring and forecasting. It will cover two bands in the LWIR and MWIR, with a remarkable spectral sampling (about
$0.603 \, \mathrm{cm}^{-1}$ and $0.604 \, \mathrm{cm}^{-1}$, respectively), combined with a high frequency of spectra acquisition (every 30 minutes over
Europe) of an entire Earth disc from a geostationary platform.

The purpose of the work was, therefore, to assess the contribution that the assimilation of radiances that IRS will acquire,
could bring to the characterisation of the atmospheric chemical composition (focus on the ozone) over Europe when assimilated
into a chemistry transport model (CTM) such as MOCAGE.





In order to fulfil these purposes, realistic data of the instrument had to be simulated. The method chosen was the Observing System Simulation Experiment (OSSE). A detailed study was carried out in order to build a robust OSSE framework over the European domain. For both the Nature Run (NR - i.e. the reference reality) and the Control Run (CR), the MOCAGE model

was chosen. This provides a global (GLOB11) and regional (MACC01) domain configuration. To have the regional domain, i.e. the one of interest for the project, the global has to run too, since MOCAGE is a nested grid chemistry transport model. The research configuration with 60 vertical levels, up to 0.1 hPa, has been exploited. The time-period chosen for the evaluation of the simulations ranged between the 1st of June till the 31st of August, 2019.

Planning for the construction of a realistic NR to be different enough from the CR, but consistent with it, required a significant

research effort. For both runs, the MOCAGE model, forced by the same meteorological input from the operational ARPEGE, has been used. Therefore, a strategy had to be chosen to differentiate the runs. First, different emission configurations, referring to different years, have been set for the NR and CR frameworks. Plus, another, more unconventional, choice was made in order to differentiate the reference reality and the run of control: L1c radiances from the IASI instrument were assimilated within the global MOCAGE configuration in the NR framework. The MACC01 domain was then used as NR, which is forced at the

edges by the global domain thus obtained, but within which, in turn, no assimilation was performed. In this case the ability of MOCAGE to benefit from IRS observations is studied.

A study of inter-comparison has, then, been performed between NR and CR (which we remind to be the simulations on the regional MACC01 MOCAGE domain). The noticeable differences that emerged have been estimated enough to avoid the identical twin problem. The $O_x$ total columns averaged over the time-frame considered for the evaluation (from 1st of June to

31st of August, 2019), showed a CR producing always lower values than the NR, with a relative difference between 10.5 and 13.5%. The spatial distribution of the values was consistent for the two runs, as were the errors.

From the NR, IRS observations have been simulated, using RTTOV v12, for the 195 contiguous wavelengths bounded in 982.464 and 1099.467 cm$^{-1}$, i.e. the range that had been confirmed by *a priori* sensitivity studies (not shown) to be the most sensitive to the ozone species and, then, the most suitable for the purposes of this work.

Only clear-sky observations have been simulated (cloud filter applied through information issued from the meteorological forcing). A horizontal thinning of the observation was also applied in order to make the ensuing assimilation reliable and efficient in terms of computational time (one pixel simulated per box of $0.4°$). These actions have produced a good observation density and frequency along the time period of interest.

Perfect observations thus created, have been perturbed with the addition of the IRS instrument noise provided by EUMET-

SAT.

The synthetic observations have been assimilated within the CR using a 3D-Var method with an hourly assimilation window. Background errors have been derived as a percentage of the ozone profile (2% over the entire column). Observation errors, instead, have been obtained by means of the so-called *Desroziers diagnostics*.

Statistics of residuals and innovations have been computed and verified, and found to be correct, i.e. analysis closer to

observations than the background. Furthermore, higher values have been found at the edges of the domain than in the inner area, in link with the coupling information coming from the lateral boundary conditions.




The contribution of IRS synthetic radiance assimilation has been assessed.

A study on the total columns of ozone averaged over the whole period of study showed that the assimilation of IRS radiances into the MOCAGE model always has a positive impact compared to no assimilation. The evaluated total columns averages, obtained from the AR of IRS, deviated from NR very little where synthetic observations are more frequently assimilated. Values were mostly close to $\sim 0\%$, with just a few areas touching $\sim 3\%$. This has represented a significant improvement over the CR case, which, instead, deviated from the NR up to $\sim 13\%$. Errors resulted spatially consistent and of comparable magnitude. Standard deviation of the bias existing between AR of IRS with respect to the NR, had its smallest values in the area where IRS radiances have been assimilated ($\sim 1.8\,\mathrm{DU}$).

When evaluating tropospheric columns, slightly stronger variations compared to the case of the total columns have been estimated, with values closer to $3\%$ than to zero.

Comparing with what obtained for the CR, where vertical variations were much more evident (up to the $25\%$), the impact of assimilation is considerable on the whole vertical column.

From the results obtained and examined, many perspectives open up.

The first thing to do in the very short term is to improve the assessment of the assimilation. It has been stated that the impact of coupling between MOCAGE domains produces more variable values of innovations and, thus, of ozone field at the southern edge of the treated MACC01 regional domain. To ensure that the averages over the domain are not impacted by these values, the impact of assimilation should also be evaluated on a smaller area, that cuts off these edges.

It is official that Principal Components (PCs) only will be distributed for IRS. The present study, however, did not take into account this subject. A short study evaluating the different impact of PCs on the work illustrated so far, should be carried out. Of course, this would be possible, at this stage, for the official version of IRS only (and not for IRS*2).

Another point that arose from the study on the preparation of the NR framework has been the possibility to perform a channel selection, while maintaining a good quality in the assimilation results. This is also a commonly used procedure in NWP for many instruments, that has been explored at CNRM for IRS [Coopmann et al. (2022)] and in the context of another future hyperspectral infrared instrument IASI-NG (IASI new generation) [Vittorioso et al. (2021)]. Since the work carried out here on IRS was a first analysis, we wanted to investigate the behaviour of all the consecutive wavelengths considered. Moreover, using the full spectral band was a fair way of comparing the impact of IRS and IRS*2. At a later stage, however, it will be of interest to carry out a selection of a smaller group of channels too.

If we consider assimilating IRS in the global domain of MOCAGE, the joint assimilation of IRS with GIIRS (hyperspectral IR sounder on board chinese geostationary satellites) could lead to interesting validation. At the same time, on board the MTG-S satellites, the Ultraviolet Visible Near-Infrared imaging spectrometer Sentinel-4 will also fly close to IRS. This is designed to monitor some key air quality trace gases and aerosols over Europe with a high spatial resolution and fast revisit time, in support of CAMS. Given the potential of the joint acquisition of the two instruments, their synergy in the characterisation of atmospheric pollution over Europe should definitely be tested.

The joint assimilation of IRS radiances with data acquired by other spectrometers working in a similar way, though on polar-orbiting platforms, would also be interesting to be considered. First on the list are certainly IASI and IASI-NG (as soon

as its data are exploitable), on board European satellites. Other IR sounders of the US and Chinese polar systems (CrIS and HIRAS) could be added also.

Since IRS provides good vertical information in stratosphere, UTLS and free troposphere, surface data could also bring
additional details at the lower-most atmospheric layers and then be assimilated as a complement.

Finally, to be certainly addressed as a continuation of this work, there will be the evaluation of the assimilation of radiances in the bands sensitive to other chemical species and aerosols. The OSSE framework created for this study was designed to provide, with minor modifications the basics for the extension to other species than ozone. The sulphur dioxide ($SO_2$) could be investigated since IRS has the potential to sense this species, with a sensitivity towards the end of MWIR band [Coopmann
et al. (2022)]. More specific case study, however, should be taken into account for this compound, since it is associated to specific and punctual natural events. The carbon monoxide (CO) is, however, going to be the first target, given the results encountered during the study performed to assess the sensitivity of IRS in its two versions.

*Code availability.*  The code used to generate the analysis (MOCAGE and its variational assimilation suite) is a research-operational code property of Météo France and CERFACS and is not publicly available yet. The readers interested in obtaining parts of the code for research
purposes can contact the authors of this study directly.

*Data availability.*  All results are available upon request to the author.

**Appendix A:  Pressure Equivalence for MOCAGE Vertical Levels**

*Author contributions.*  FV, VG and NF designed the analysis and FV carried out and interpreted the simulations. FV is the main contributor to the manuscript, VG and NF reviewed and contributed to the manuscript. NF and VG secured the funding.

*Competing interests.*  No competing interest.

*Acknowledgements.*  This research has been carried out in the framework of a PhD project co-funded by Météo-France and Thales Alenia Space. We really want to express our gratitude to Dr. Marine Claeyman and Francis Olivier for their scientific advises and guidance during the doctoral project.





| Level | Pressure Equivalence (hPa) | | | Level | Pressure Equivalence (hPa) | | |
|---|---|---|---|---|---|---|---|
| | Min. | Median | Max | | Min. | Median | Max |
| 1 | 0.1 | 0.1 | 0.1 | 31 | 85.4 | 106.4 | 109.1 |
| 2 | 0.3 | 0.3 | 0.3 | 32 | 95.2 | 120.4 | 123.6 |
| 3 | 0.5 | 0.5 | 0.5 | 33 | 106.1 | 136.3 | 140.1 |
| 4 | 0.8 | 0.8 | 0.8 | 34 | 118.3 | 154.3 | 158.9 |
| 5 | 1.1 | 1.1 | 1.1 | 35 | 131.9 | 174.7 | 180.1 |
| 6 | 1.6 | 1.6 | 1.6 | 36 | 147 | 197.7 | 204.2 |
| 7 | 2.1 | 2.1 | 2.1 | 37 | 163.9 | 223.8 | 231.3 |
| 8 | 2.7 | 2.7 | 2.7 | 38 | 182.6 | 252.9 | 261.9 |
| 9 | 3.4 | 3.4 | 3.4 | 39 | 203.0 | 285.2 | 295.6 |
| 10 | 4.4 | 4.4 | 4.4 | 40 | 225.2 | 320.7 | 332.8 |
| 11 | 5.6 | 5.6 | 5.6 | 41 | 249.4 | 359 | 373.6 |
| 12 | 7.0 | 7.0 | 7.0 | 42 | 275.5 | 402.4 | 418.4 |
| 13 | 8.6 | 8.7 | 8.7 | 43 | 302.9 | 447.4 | 465.7 |
| 14 | 10.5 | 10.6 | 10.7 | 44 | 330.1 | 492.7 | 513.3 |
| 15 | 12.5 | 12.8 | 12.8 | 45 | 357.3 | 538.1 | 561.0 |
| 16 | 14.8 | 15.2 | 15.3 | 46 | 384.0 | 583.2 | 608.5 |
| 17 | 17.2 | 17.9 | 18.0 | 47 | 410.2 | 627.7 | 655.2 |
| 18 | 19.7 | 20.8 | 20.8 | 48 | 435.8 | 671.2 | 701.1 |
| 19 | 22.5 | 23.9 | 24.1 | 49 | 460.5 | 713.5 | 745.6 |
| 20 | 25.5 | 27.2 | 27.5 | 50 | 484.2 | 754.3 | 788.5 |
| 21 | 28.6 | 30.9 | 31.2 | 51 | 506.7 | 793.1 | 829.4 |
| 22 | 31.9 | 34.9 | 35.3 | 52 | 527.9 | 829.7 | 868.0 |
| 23 | 35.5 | 39.4 | 39.9 | 53 | 547.5 | 863.8 | 903.9 |
| 24 | 39.6 | 44.6 | 45.2 | 54 | 565.5 | 895.0 | 936.8 |
| 25 | 44.3 | 50.4 | 51.2 | 55 | 581.6 | 923.1 | 966.3 |
| 26 | 49.4 | 57.1 | 58.1 | 56 | 595.8 | 947.6 | 992.2 |
| 27 | 55.2 | 64.7 | 65.9 | 57 | 607.7 | 968.3 | 1014.0 |
| 28 | 61.6 | 73.3 | 74.8 | 58 | 617.3 | 984.9 | 1031.5 |
| 29 | 68.7 | 83.0 | 84.9 | 59 | 624.4 | 997.1 | 1044.3 |
| 30 | 76.6 | 94.1 | 96.3 | 60 | 628.9 | 1004.4 | 1052.0 |

**Table A1.** Minimum, median and maximum value of pressure encountered over the domain MACC01 (28°N, 26°W and 72°N, 46°E) for each of the 60 MOCAGE vertical levels.



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
