# Peer review of "Assessment of the contribution of IRS for the characterisation of ozone over Europe"

_Atmospheric Measurement Techniques, 2024_

## Referee Comment (RC1)

**Review of the manuscript "Assessment of the contribution of IRS for the characterisation of ozone over Europe" by Vittorioso et al.**

This paper reports the outcome of an observing system simulation experiment (OSSE) conducted with simulated data from a future geostationary sensor, the MTG – Infrared Sounder (IRS). Only ozone sensitive channels are simulated and assimilated. The authors demonstrate that the IRS can be expected to have a positive impact on assimilated total ozone over Europe. The impacts on ozone profiles are less clear and, in my opinion, could use more discussion (see my general comment). Production and evaluation of an OSSE is an important step in the development of a new sensor. As such this work is firmly within the scope of AMT. Overall the manuscript is easy to follow and well sourced although the writing style could use some polishing. I've made some suggestions to that effect in my technical comments.

**General comment**

I would like to see some discussion of why assimilation of IRS radiances improves the mean and standard deviation of total and tropospheric ozone columns but not the profiles of standard deviation of the difference with the nature run (Fig. 14b). One possibility is that there's very little profile information in those radiances so that the increments simply reflect corrections to the total column and how these increments are distributed in the vertical is determined by a combination of the prescribed B matrix and altitudes where the weighting functions for those channels happen to peak. The resulting analysis profile may then have little to do with the "correct" one and instead represent the best fit to total ozone given the constraints. It may be instructive to see a plot of the weighting functions for the selected channels or some other metric of the sensitivity of those radiances to ozone distributions in the vertical. This is not a criticism of this work. The results are what they are and if IRS measurements alone cannot constrain ozone profiles, then this is an important conclusion, which the paper should clearly communicate and explain why this is the case. Generally speaking, it's hard to constrain profiles using nadir data so that conclusion would not be surprising. On the other hand, perhaps a different channel selection would do a better job?

**Specific comments**

L15-17. What about the representation of ozone variability? Standard deviations? Correlations?

L22 What behavior?

L25. I think the preferred term is "observing system".

L44. For those of us (myself included) less familiar with orbital dynamics, what does it mean for an orbit to be "*wider*"?

L106. Presumably not **all** the information content will be preserved in this dimension reduction procedure. Vast majority of it?

LL106-107. I would prefer to see an explicit statement of what approach is taken here instead of the PCs. Straightforward channel selection, presumably?

LL162-163. I suggest more careful wording here. This sentence makes it sound like the compromise is between observations and some kind of climate simulation. But that's not the case. The background state itself is obtained from a multitude of past observations previously assimilated and propagated by the model. I think this is an important point that is so often missed in casual explanations of DA. It's a very much data-based methodology.

L211. But in this case it's not a free-running simulation, is it? It is a specified dynamics simulation and it assimilates IASI. It would be good either to say it here or to drop this bit from the sentence. This sentence confused me a lot when I was trying to understand the OSSE setup in the next section.

L241-243. This sentence could be significantly shortened by omitting the explanation why the NR and CR must be different. This explanation has been given twice already. More generally, I suggest streamlining this and the preceding paragraph somewhat better to make it follow the structure of Fig.5 more closely. First, I would say that both the NR and CR use specified dynamics from ARPEGE and use the same CTM. Then I would say that two modifications were introduced to differentiate the CR from the NR and describe what they are: (1) emissions; (2) IASI assimilation in the NR.

L316. What was used for the noise and how the appropriate amount of noise was determined? Is it a realistic representation of the expected instrumental noise?

L341. "*higher values and most significant spread from the instrument noise*". I'm struggling to follow. Do you mean higher values and a greater spread than that seen in the instrumental noise?

L342. I don't understand what is meant by "*the small error in the observation operator*" and why it is responsible for the estimated uncertainties being larger than the noise added to the simulated observations. Perhaps you mean sources of error other than instrumental noise such as pointing error, representativeness error etc.?

L344. This sentence is tautological: correlations are related to covariance by construction. I suggest deleting it. It would, however, be interesting to understand how the correlations arise. Could you briefly comment on this?

Figure 11 and discussion. What exactly is plotted there? B is a covariance matrix so the units should be mixing ratio squared but the figure says ppmv. Is it standard deviations, i.e. square root of the diagonal?

Figure 13 should use the same color schemes and ranges as those in Fig. 6 for the reader to see how well the AR compares to the NR relative to the CR.

L384 and 393. I find the term "*error of the bias*" confusing (the bias is itself the average error). I would just call it standard deviation of the differences as you do in the caption. Similarly in the

discussion of Fig. 6 I saw the term "*standard deviation of the bias*". Same comment: it should be standard deviation of the difference.

L394. I think you mean "*color scheme*", not "*color map*".

L410. What is "*model restitution*"?

LL406-411. See my general comment above. The standard deviation result is disappointing but not necessarily unexpected. I think it merits more discussion. For example, could it be because there is very little vertical information in the radiance data and assimilation thus distributes increments incorrectly in the vertical? Plotting a few weighting functions might help.

Data availability. Ultimately it's up to the editor but I don't think "*All results are available upon request to the author*" is sufficient to meet the Copernicus requirements. At least the data used to make the figures and statistics should be in a publicly accessible repository. See the data policy here: https://www.atmospheric-measurement-techniques.net/policies/data_policy.html

**Technical comments**

L13 "along this study was the ozone" → in this study was ozone.

LL28-29. "they do not exist everywhere, since huge gaps exist between weather stations." Suggestion: they are spatially sparse.

L35. "within" → between

L66 "Since this study is a preliminary study" → Since this is a preliminary study.

L67. "the ozone" → ozone

L74 "oxidant" → oxidizing power

L81. "are issued from" → are based on

L118. "precious" → valuable

L121. "disposes of". Do you mean "consists of" or "allows"? In any case, you don't want to say "disposes of", which means "throws away".

L125-127. The sentence starting with "Through this system…" is very unclear. I actually don't understand it at all. Please, rephrase.

L203. "comply"; I think you mean "consist of" or something like that.

L213. Please revise this sentence for grammar and structure.

L251. Instead of saying "in the following", please specify in which subsection these details are provided.

Table 1. "Anthropic" → Anthropogenic.

L275. "in troposphere" → in the troposphere.

L292. What does "stronger" mean? Greater than?

L297. "sub-set" → subset

L342 "zone" → ozone

L347. "strongest values" → largest values.

L402. 'weaker' → lower.

L456. "in link". I think this should be "consistent with" or something like that.

The title of Appendix A appears to be in the wrong place. It is followed by statements on CI and acknowledgments and then a table, which I think is supposed to be the (sole?) content of the Appendix.

Thanks,
Kris Wargan

---

## Author Comment (AC1)

**Reviewer 2 - Timothy Schmit**

We would like to sincerely thank the Reviewer for taking the time to read and review the manuscript.

**general comments**

Much work went into this simulation study to develop a basis for the potential impact on ozone from EUMETSAT's high spectral IR sounder. Overall, the manuscript is a great first step in showing the benefits of geostationary advanced sounders regarding Ozone, and a pathway for investigating other species. Seems like the manuscript script could have been much shorter if the authors had used different models for the simulations and assimilation steps. For example, maybe http://raqms-ops.ssec.wisc.edu/, but this was not the path taken, so much on the calibration of the OSSE had to be included.

We thank the Reviewer for this remark. This has been a very preliminary and pioneering study about assimilation of L1 simulated radiances from a new infrared instrument into a Chemistry Transport Model, and the resources available have been exploited. Indeed, this implied that the approach chosen was to use the same model for both Nature Run and Control Run, acting on the sources of differences between them. A more traditional approach could be taken in possible future studies.

**specific comments**

- For more on the instrument, this reference on the IRS could be added: Sylvain Abdon, Hubert Gardette, Cyril Degrelle, Jean-Michel Gaucel, Patrick Astruc, Patrice Guiard, Antonio Accettura, Daniel Lamarre, Donny M. Aminou, Didier Miras, "Meteosat third generation infrared sounder (MTG-IRS), interferometer and spectrometer test outcomes, demonstration of the new 3D metrology system efficiency," Proc. SPIE 11852, International Conference on Space Optics — ICSO 2020, 118521F (11 June 2021); https://doi.org/10.1117/12.2599240
  Reference suggested has been cited.

- Line 8. Since the IRS uses the IR part of the EM, the various gases can be monitored at night as well as during the day. This fact should be note somewhere before the summary. Stated another way, advanced IR sounders in geo can complement UV/visible sensors. Maybe add this reference: https://journals.ametsoc.org/view/journals/bams/104/3/BAMS-D-22-0266.1.xml
  Thanks for this complement, the following paragraph has been added in the Introduction:
  "Since covering a part of the infrared portion of the electromagnetic spectrum, IRS will be able to provide both day and nighttime data. This could potentially complement the information issued from UV/VIS sensors currently used for monitoring from GEO platforms [Kopacz et al. (2023)]".

- Line 15. Define that NR is the Natural Run. Currently defined on line 208.
  Comment taken into account.

- Line 20. This statement would be equally true in operations, not just in research.
  Text has been modified.

- Line 57. Could include this overview article on the MTG: Holmlund, K., and Coauthors, 2021: Meteosat Third Generation (MTG): Continuation and Innovation of

Observations from Geostationary Orbit. *Bull. Amer. Meteor. Soc.*, **102**, E990–E1015,
https://doi.org/10.1175/BAMS-D-19-0304.1.
Reference suggested has been cited.

- Line 318. What local zenith angle cut-off was used?
  We used a cut-off for satellite zenith angles at 63 degrees.

- Line 350. The reader could be reminded that the IRS will be providing data over Europe every 30 min, while only hourly data were included in this study.
  Indeed, it has not been mentioned in the manuscript that the observations were simulated on an hourly basis. We have added the following paragraph in Subsection 5.3:
  "Although the IRS instrument will be capable of providing an acquisition every 30 minutes over LAC4 (as introduced in Section 2), in this work only one set of observations per hour has been simulated. This choice has the additional aim of further optimising the subsequent assimilation and reducing the computational cost."

- Line 335. As a start, only clear sky values were used, but for low clouds, won't the sensor still observe most of the ozone?
  Indeed, most of the ozone signal can be seen by IRS. However, if we want to use this information, we first have to identify the channels contaminated by the low cloud. This would need a cloud detection scheme (like the McNally and Watts algorithm). Unfortunately, we have not implemented yet this algorithm in MOCAGE. So we may underestimate the benefit of IRS in our study because we only use clear scenes.

**technical corrections**

- Line 27. Replace trough with through.
  Text has been modified as suggested.

- Line 37. Consider changing to "wide spectral range".
  Text has been modified as suggested

- Line 203. This kind of experiments comply a series of steps should be These kind of experiments comply a series of steps
  Modified to "Such experiments"

- Line 247. Remove 'it' … At the same time, it the same surface emissions as ….
  Comment taken into account.

- Line 219. The text states "observations are more dense over land and in the South-East portion of the instrument disk", yet the figure (8) is mostly just covering Europe.
  Text has been changed to: "South-East quadrant of the domain".

---

## Author Comment (AC2)

Review of the manuscript "Assessment of the contribution of IRS for the characterisation of ozone over Europe" by Vittorioso et al.

This paper reports the outcome of an observing system simulation experiment (OSSE) conducted with simulated data from a future geostationary sensor, the MTG – Infrared Sounder (IRS). Only ozone sensitive channels are simulated and assimilated. The authors demonstrate that the IRS can be expected to have a positive impact on assimilated total ozone over Europe. The impacts on ozone profiles are less clear and, in my opinion, could use more discussion (see my general comment). Production and evaluation of an OSSE is an important step in the development of a new sensor. As such this work is firmly within the scope of AMT. Overall the manuscript is easy to follow and well sourced although the writing style could use some polishing. I've made some suggestions to that effect in my technical comments.

We would like to sincerely thank the reviewer for taking the time to read this paper and review it.

**General comment**

I would like to see some discussion of why assimilation of IRS radiances improves the mean and standard deviation of total and tropospheric ozone columns but not the profiles of standard deviation of the difference with the nature run (Fig. 14b). One possibility is that there's very little profile information in those radiances so that the increments simply reflect corrections to the total column and how these increments are distributed in the vertical is determined by a combination of the prescribed B matrix and altitudes where the weighting functions for those channels happen to peak. The resulting analysis profile may then have little to do with the "correct" one and instead represent the best fit to total ozone given the constraints. It may be instructive to see a plot of the weighting functions for the selected channels or some other metric of the sensitivity of those radiances to ozone distributions in the vertical. This is not a criticism of this work. The results are what they are and if IRS measurements alone cannot constrain ozone profiles, then this is an important conclusion, which the paper should clearly communicate and explain why this is the case. Generally speaking, it's hard to constrain profiles using nadir data so that conclusion would not be surprising. On the other hand, perhaps a different channel selection would do a better job?

Assimilation of IRS radiances mainly improves the bias (as seen in Figure 14a of the manuscript), while it improves only to a smaller extent, or sometimes deteriorates, standard deviation of differences to the Nature Run (Figure 14b of the manuscript).
When looking at the Jacobian values (see Figure R1 - left), one can observe a strong sensitivity to an atmospheric layer ranging from model level 40 to 45 (i.e. 320 to 538 hPa). When switching to a representation of the Jacobians multiplied by 10% of the ozone profile itself, the maximum sensitivity moves up to a layer found between model level 20 and 25 (i.e. 27-50 hPa). As a consequence, the information is not retrieved along the whole vertical column. It mainly comes from very specific atmospheric layers. Nevertheless, roughly two groups of channels can be identified plotting Jacobians (see better in R2): one sounding Stratosphere and the other sounding the Troposphere.

When looking at studies performed on IRS bands information content (Coopmann et al 2022) one can easily deduce, indeed, that very little profile information is found in these kinds of simulated radiances. Coopmann et al. (2022) report a DFS value for ozone that goes from a maximum of ~7 for the entire IRS spectrum, while reducing up to ~2 when performing channel selections and then reducing the channel amount. Their study used a different B matrix than our work, which has an impact on the DFS value. Going beyond the value itself, they also found two major locations for improvement: UTLS and troposphere.

We cannot make a strong conclusion on the real capacity of IRS to modify the entire ozone profile, until further studies are made using more refined description of the B matrix, for instance. Such work is envisioned in the Horizon project CAMEO (task 3.3.2, https://www.cameo-project.eu/). Joint assimilation of UV-VIS and IR could also be studied (like IRS and Sentinel 4).

1. Coopmann, O., Fourrié, N., and Guidard, V.: Analysis of MTG-IRS observations and general channel selection for numerical weather prediction models, Quarterly Journal of the Royal Meteorological Society, 2022.

[Figure]

Figure R1: Ozone Jacobians for the 195 channels simulated for IRS (a), both simple (left) and normalised to the 10% of the ozone profile itself (right).

[Figure]

Figure R2: another representation of R1

**Specific comments**

- L15-17. What about the representation of ozone variability? Standard deviations? Correlations?

  Indeed, we propose to modify the last paragraph of the Summary as follows:

  "The results obtained indicate that the assimilation of synthetic radiances of IRS always has a positive impact on the ozone analysis from the model MOCAGE. The relative average difference compared to the Nature Run (NR) in the ozone total columns improves from -30% (no assimilation) to almost zero when IRS observations are available over the domain. Remarkable is also the reduction in the standard deviation of the difference with respect to the NR, which, in the area where IRS radiances are assimilated, reaches its lowest values (~ 1.8 DU).
  When considering tropospheric columns the improvement is also significant, from 15-20% (no assimilation) down to 3%. The error of the differences compared to NR is lower than for total columns (minima ~ 0.3 DU), due also to the lower concentrations of the tropospheric ozone field. Overall, the impact of assimilation is considerable over the whole vertical column: vertical variations are noticeably improved compared to what obtained when no assimilation is performed (up to 25% better)."

- L22 What behavior?

  Text modified to: "evolution of atmospheric chemical state"

- L25. I think the preferred term is "observing system".
  The text has been modified

- L44. For those of us (myself included) less familiar with orbital dynamics, what does it mean for an orbit to be "wider"?
  Rephrased to: "Geostationary motions require the satellite to cover an orbit much further from the Earth surface than satellites in polar orbit."

- L106. Presumably not all the information content will be preserved in this dimension reduction procedure. Vast majority of it?
  The text has been modified.

- LL106-107. I would prefer to see an explicit statement of what approach is taken here instead of the PCs. Straightforward channel selection, presumably?
  The following phrase has been added: "As better explained in the following of the paper (Subsection 5.3) we chose to work with raw radiances in the ozone band."

- LL162-163. I suggest more careful wording here. This sentence makes it sound like the compromise is between observations and some kind of climate simulation. But that's not the case.I think this is an important point that is so often missed in casual explanations of DA. It's a very much data-based methodology. The background state itself is obtained from a multitude of past observations previously assimilated and propagated by the model.
  Added in the text after the Equation: "In the present case xb is obtained from a forecast from the previous assimilation."

- L211. But in this case it's not a free-running simulation, is it? It is a specified dynamics simulation and it assimilates IASI.
  Correct. As explained in the next following subsection, the assimilation is performed only in the Global configuration of the model. This latter forces the regional model state through the boundary conditions.
  It would be good either to say it here or to drop this bit from the sentence.
  This sentence confused me a lot when I was trying to understand the OSSE setup in the next section.
  This text has been added: "In the present study the approach is more elaborated and introduced in Subsection 5.1."

- L241-243. This sentence could be significantly shortened by omitting the explanation why the NR and CR must be different. This explanation has been given twice already. More generally, I suggest streamlining this and the preceding paragraph somewhat better to make it follow the structure of Fig.5 more closely. First, I would say that both the NR and CR use specified dynamics from ARPEGE and use the same CTM. Then I would say that two modifications were introduced to differentiate the CR from the NR and describe what they are: (1) emissions; (2) IASI assimilation in the NR.
  Text has been modified to: "The first step was to differentiate the …of IRS observations" was replaced by:

"Two modifications were introduced to differentiate the CR from the NR:

1. For biogenic and anthropogenic emissions in the NR framework, the configurations used in operational MOCAGE at the time this work began, have been employed for each geographical domain. For the CR framework, on the other hand, data referring to the year 2000 were used. This provided the same kind of spatial variability for this class of emissions than the NR, but with different intensities. For the representation of the biomass burning, data from the CAMS GFAS, were used as input to MOCAGE for the NR settings, while for the CR was MACCity representative of the year 2000. For the details about the emissions provided to MOCAGE for each run and domain, see Table 1.

2. Radiances from the Metop-IASI spectrometer have been assimilated into the MOCAGE model in the GLOB11 configuration. However, the regional domain MACC01 nested in the global, and in which no assimilation was directly carried out, has been the NR employed for the simulation of IRS observations."

- L316. What was used for the noise and how the appropriate amount of noise was determined? Is it a realistic representation of the expected instrumental noise?
  The noise added here is the one depicted in Figure 3 and has been provided by EUMETSAT, from Thalès-Alenia-Space inputs. Which is indeed supposed to be realistic. A reference to Figure 3 has been added in the sentence.

- L341. "higher values and most significant spread from the instrument noise". I'm struggling to follow. Do you mean higher values and a greater spread than that seen in the instrumental noise?
  The text has been replaced by:
  "The diagnosed standard deviation shows higher values than the instrument noise. Indeed, the observation error includes the instrumental noise, the observation operator error and the representativeness error. Similarly, the observation error cross-channel correlations (Figure 10.b), which are not present in the instrumental noise, may arise from the contribution of the observation operator errors. The small error in the observation operator in the ozone treatment can explain this additional contribution to the whole observation error."

- L342. I don't understand what is meant by "the small error in the observation operator" and why it is responsible for the estimated uncertainties being larger than the noise added to the simulated observations. Perhaps you mean sources of error other than instrumental noise such as pointing error, representativeness error etc.?
  See answer to previous comment. Indeed, we were referring to the observation operator error, contributing to the whole observation error.

- L344. This sentence is tautological: correlations are related to covariance by construction. I suggest deleting it. It would, however, be interesting to understand how the correlations arise. Could you briefly comment on this?
  See answer to previous comment. The correlations are coming from the contribution of the observation operator errors. The related sentence has been kept.

- Figure 11 and discussion. What exactly is plotted there? B is a covariance matrix so the units should be mixing ratio squared but the figure says ppmv. Is it standard deviations, i.e. square root of the diagonal?
  Indeed, Figure 11 represents background error standard deviations. The correct sentence is: The B matrix was obtained as introduced in Subsection 4.2. An example of the vertical cross section (average of the longitudes) of the background error standard deviations.

- Figure 13 should use the same color schemes and ranges as those in Fig. 6 for the reader to see how well the AR compares to the NR relative to the CR.
  Indeed, for biases values are only and strongly negative for the CR, contrary to the NR where they are centered around zero. For the standard deviations, on the other hand, the same color schemes could be used.

- L384 and 393. I find the term "error of the bias" confusing (the bias is itself the average error). I would just call it standard deviation of the differences as you do in the caption. Similarly in the discussion of Fig. 6 I saw the term "standard deviation of the bias". Same comment: it should be standard deviation of the difference.
  Text has been modified as suggested.

- L394. I think you mean "color scheme", not "color map".
  Text has been modified as suggested.

- L410. What is "model restitution"?
  The sentence has been shortened.

- LL406-411. See my general comment above. The standard deviation result is disappointing but not necessarily unexpected. I think it merits more discussion. For example, could it be because there is very little vertical information in the radiance data and assimilation thus distributes increments incorrectly in the vertical? Plotting a few weighting functions might help.
  Figures shown in the answer to the General Comment of the Reviewer may be added to the paper, if necessary for the better understanding of the results, following the Editor's additional advice to come.

- Data availability. Ultimately it's up to the editor but I don't think "All results are available upon request to the author" is sufficient to meet the Copernicus requirements. At least the data used to make the figures and statistics should be in a publicly accessible repository. See the data policy here:
  https://www.atmospheric-measurement-techniques.net/policies/data_policy.html
  Thanks for putting this issue forward. We will discuss this point with the associate editor more in details.

**Technical comments**

- L13 "along this study was the ozone" à in this study was ozone.

- LL28-29. "they do not exist everywhere, since huge gaps exist between weather stations. " Suggestion: they are spatially sparse.
- L35. "within" à between
- L66 "Since this study is a preliminary study" à Since this is a preliminary study.
- L67. "the ozone" à ozone
- L74 "oxidant" à oxidizing power
- L81. "are issued from" à are based on
- L118. "precious" à valuable
- L121. "disposes of". Do you mean "consists of" or "allows"? In any case, you don't want to say "disposes of", which means "throws away".
  Text changed to: "provides"
- L125-127. The sentence starting with "Through this system…" is very unclear. I actually don't understand it at all. Please, rephrase.
- L203. "comply"; I think you mean "consist of" or something like that.
- L213. Please revise this sentence for grammar and structure.
- L251. Instead of saying "in the following", please specify in which subsection these details are provided.
- Table 1. "Anthropic" à Anthropogenic.
- L275. "in troposphere" à in the troposphere.
- L292. What does "stronger" mean? Greater than?
- L297. "sub-set" à subset
- L342 "zone" à ozone
- L347. "strongest values" à largest values.
- L402. 'weaker' à lower.
- L456. "in link". I think this should be "consistent with" or something like that.
- The title of Appendix A appears to be in the wrong place. It is followed by statements on CI and acknowledgments and then a table, which I think is supposed to be the (sole?) content of the Appendix.

All the technical comments suggested by the Reviewer have been taken into account.

Thanks,
Kris Wargan

Thanks again for your time and consideration.

---

## Author Response (AR2)

**Review by editor**

Dear authors:

The reviewers are generally satisfied with your responses. However, a few issues remain.

We really appreciate the revision work done by the editor and the two reviewers.

I agree with the reviewer 1 regarding making available at the least the total ozone data to generate the figures be made available on an archive particularly since the code is not yet available.

The data of ozone 3D fields (hourly values over the full period for Nature Run, Control Run and Assimilation Run) are provided in netcdf format from Zenodo deposit with DOI.
For the Nature Run:
- June 2019 https://zenodo.org/doi/10.5281/zenodo.12634489
- July 2019 https://zenodo.org/doi/10.5281/zenodo.12635956
- August 2019 https://zenodo.org/doi/10.5281/zenodo.12643523
For the Control Run:
- June 2019 https://zenodo.org/doi/10.5281/zenodo.12570356
- July 2019 https://zenodo.org/doi/10.5281/zenodo.12570536
- August 2019 https://zenodo.org/doi/10.5281/zenodo.12570745
For the Assimilation run:
- June 2019: https://doi.org/10.5281/zenodo.12547820
- July 2019: https://doi.org/10.5281/zenodo.12565863
- August 2019: https://doi.org/10.5281/zenodo.12567704

The Data availability section has been modified accordingly.

Regarding the addition of figures by Rev. 1, I agree that this will enhance the paper. I prefer the R2 set as this is what is more commonly shown.
Some additional text will be needed and I do not understand the one weighting function which is so different from the rest, so that may need some explanation.

While investigating this behavior, we have identified that the average values displayed on figure R2 were affected by only a dozen of profiles over the domain (fig. RR1). After liaising with the RTTOV infrared development team on that behavior, we have been told that RTTOV struggles in the simulation of Jacobian computation for a few channels (Jérôme Vidot, pers.comm.). After discarding those profiles, the average Jacobians have a more regular behavior. We have not further investigated why these profiles were causing troubles.

[Figure]

Fig. RR1: Location of outlier profiles that cause problems in channel 642 jacobian computation.

Figure 10 has been added (in between figures formerly named Fig9 and Fig10) and commented through the text that follows:

"Finally, the simulated ozone Jacobians, averaged over the regional domain, are shown in Figure 10. Both simple Jacobians and Jacobians times 10% of the ozone profile itself are illustrated. In the first case, a strong sensitivity to an atmospheric layer ranging from model level 40 to 45 (i.e. 320 to 538 hPa) can be observed. On the other hand, when switching to a representation of the Jacobians multiplied by 10% of the ozone profile itself, the maximum sensitivity moves up to a layer found between model level 20 and 25 (i.e. 27-50 hPa). Negative values of sensitivity are also found between level 10 and 5, 5 to 1 hPa, i.e. in stratosphere.

As a consequence, the information is retrieved along the whole vertical column. It mainly comes from very specific atmospheric layers. Nevertheless, roughly two groups of channels can be identified: one sounding stratosphere and the other sounding the troposphere."